# A Parametric Contextual Online Learning Theory of Brokerage

**François Bachoc** [1]   **Tommaso Cesari** [2]   **Roberto Colomboni** [3][4]

## Abstract

We study the role of contextual information in the online learning problem of brokerage between traders. In this sequential problem, at each time step, two traders arrive with secret valuations about an asset they wish to trade. The learner (a broker) suggests a trading (or brokerage) price based on contextual data about the asset and the market conditions. Then, the traders reveal their willingness to buy or sell based on whether their valuations are higher or lower than the brokerage price. A trade occurs if one of the two traders decides to buy and the other to sell, i.e., if the broker's proposed price falls between the smallest and the largest of their two valuations. We design algorithms for this problem and prove optimal theoretical regret guarantees under various standard assumptions.

## 1. Introduction

Inspired by a recent stream of literature (Cesa-Bianchi et al., 2021; Azar et al., 2022; Cesa-Bianchi et al., 2024b; 2023; Bolić et al., 2024; Bernasconi et al., 2024; Bachoc et al., 2024), we approach the bilateral trade problem of brokerage between traders through the lens of online learning. When viewed from a regret minimization perspective, bilateral trade has been explored over rounds of seller/buyer interactions with a broker with no prior knowledge of their private valuations. Similarly to Bolić et al. (2024), we focus on the case where traders are willing to either buy or sell (possibly *short*; see Section 1.1), depending on whether their valuations for the asset being traded are above or below the brokerage price.

This setting is especially relevant for over-the-counter (OTC) markets. Serving as alternatives to conventional exchanges, OTC markets operate in a decentralized manner and are a vital part of the global financial system.[1] In contrast to centralized exchanges, the lack of strict protocols and regulations delegates to brokers the responsibility of bridging the gap between buyers and sellers, who may not have direct access to one another. In addition to facilitating interactions between parties, brokers leverage their contextual knowledge and market insights to determine appropriate pricing for assets. By examining factors such as supply and demand, market trends, and other asset-specific information, brokers aim to propose prices that reflect the true market value of the asset being traded. This price discovery process is a crucial aspect of a broker's role, as it helps ensure efficient transactions by accounting for the unique circumstances surrounding each asset. Additionally, in many OTC markets, as in our setting, traders choose to either buy or sell depending on the contingent market conditions (Sherstyuk et al., 2020). This behavior is observed across a broad range of asset trades, including stocks, derivatives, art, collectibles, precious metals and minerals, energy commodities like gas and oil, and digital currencies (cryptocurrencies).

We propose a *contextual* version of the online brokerage problem, that is of significant practical interest given that the broker often has access to meaningful information about the asset being traded and the surrounding market conditions *before* having to propose a trading price. This information might help the broker to propose more targeted trading prices by inferring (an approximation of) the current market value of the corresponding asset, and ignoring it could be extremely costly in terms of missing trading opportunities.

Although an extensive amount of work has been done on non-contextual bilateral trade problems (including brokerage problems), the existing literature on the more realistic contextual versions of these problems is scarce (see Section 1.3). The main reason for the slower development of contextual results is the higher complexity of these settings and the impossibility of simply adapting non-contextual al-

[1]IMT and IUF, Université Paul Sabatier, Toulouse, France [2]EECS, University of Ottawa, Ottawa, Canada [3]DEIB, Politecnico di Milano, Milano, Italy [4]Department of CS, Università degli Studi di Milano, Milano, Italy. Correspondence to: François Bachoc <francois.bachoc@math.univ-toulouse.fr>, Tommaso Cesari <tcesari@uottawa.ca>, Roberto Colomboni <roberto.colomboni@polimi.it>.

*Proceedings of the $42^{nd}$ International Conference on Machine Learning*, Vancouver, Canada. PMLR 267, 2025. Copyright 2025 by the author(s).

---

[1]In the US alone, the value of assets traded in OTC markets exceeded a remarkable 50 trillion USD in 2020, surpassing centralized markets by more than 20 trillion USD (Weill, 2020). This growth has been steadily increasing since 2016 (www.bis.org, 2022).

gorithmic ideas and analyses to their contextual counterparts. We aim to fill this gap in the online learning literature on bilateral trade to guide brokers in these contextual scenarios.

### 1.1. Setting

In the following, the elements of any Euclidean space are treated as column vectors and, for any real number $x, y$, we denote their minimum by $x \wedge y$ and their maximum by $x \vee y$.

**Online protocol.** We study the following problem.

At each time $t \in \mathbb{N}$,

- ○ Two traders arrive with private valuations $V_t, W_t \in [0, 1]$ about an asset they want to trade.

- ○ The broker observes a context $c_t \in [0, 1]^d$ and proposes a trading price $P_t \in [0, 1]$.

- ○ The two bits $\mathbb{I}\{P_t \le V_t\}, \mathbb{I}\{P_t \le W_t\}$ (i.e., the willingness of each trader to buy or sell) are revealed to the broker.

- ○ If the price $P_t$ lies between the lowest valuation $V_t \wedge W_t$ and highest valuation $V_t \vee W_t$ (meaning the trader with the minimum valuation is ready to sell[2] at $P_t$ and the trader with the maximum valuation is eager to buy at $P_t$), the asset is bought by the trader with the highest valuation from the trader with the lowest valuation at the brokerage price $P_t$.

**Market value.** At any time $t \in \mathbb{N}$, the context $c_t$ is related to the traders' valuations $V_t, W_t$ via the hidden *market value* of the asset: a number $m_t \in [0, 1]$ that satisfies the two assumptions below, which are assumed to hold throughout the whole paper.

The first assumption (Assumption 1.1) states that an unknown linear relation exists between the unknown market value $m_t$ and the corresponding context $c_t$ the broker observes before proposing a trading price.

**Assumption 1.1** (Market values and contexts)**.** There exists $\phi \in [0, 1]^d$, unknown to the broker, such that, for each $t \in \mathbb{N}$, it holds that $m_t = c_t^\top \phi$.

The second assumption accounts for variability due to personal preferences or individual needs of the traders by modeling traders' valuations as zero-mean perturbations of market values.

**Assumption 1.2** (Market values and valuations)**.** There exists an independent sequence of random variables

$\xi_1, \zeta_1, \xi_2, \zeta_2, \dots$ such that, for each $t \in \mathbb{N}$, it holds that $\mathbb{E}[\xi_t] = 0 = \mathbb{E}[\zeta_t]$ and $V_t = m_t + \xi_t$ and $W_t = m_t + \zeta_t$.[3]

**Contexts.** We model the sequence of contexts $c_1, c_2, \dots$ as a deterministic $[0, 1]^d$-valued sequence (possibly generated by an adversarial environment with knowledge of the broker's algorithm) that is initially unknown but sequentially discovered by the broker. As a consequence, note that the sequence of market values $m_1, m_2, \dots$ can change arbitrarily (and even adversarially) from one time step to the next.[4]

**Gain from trade and Regret.** Consistently with the existing bilateral trade literature, the reward associated with each interaction is the sum of the net utilities of the traders, known as *gain from trade*. Formally, for any $p, v, w \in [0, 1]$, the utility of a price $p$ when the valuations of the traders are $v$ and $w$ is

$$\mathrm{g}(p, v, w) \coloneqq ( \underbrace{v \vee w - p}_{\text{buyer's net gain}} + \underbrace{p - v \wedge w}_{\text{seller's net gain}} )\underbrace{\mathbb{I}\{v \wedge w \le p \le v \vee w\}}_{\text{a trade happens}}$$

$$= (v \vee w - v \wedge w)\,\mathbb{I}\{v \wedge w \le p \le v \vee w\}.$$

The aim of the learner is to minimize the *regret* with respect to the best *not-necessarily-linear* function of the contexts,[5] defined, for any time horizon $T \in \mathbb{N}$, as

$$R_T \coloneqq \sup_{p^\star : [0,1]^d \to [0,1]} \mathbb{E}\left[\sum_{t=1}^{T}\Big(\mathrm{GFT}_t\big(p^\star(c_t)\big) - \mathrm{GFT}_t(P_t)\Big)\right],$$

where we let $\mathrm{GFT}_t(p) \coloneqq \mathrm{g}(p, V_t, W_t)$ for all $p \in [0, 1]$, and the expectation is taken with respect to the randomness in $(\xi_t, \zeta_t)_{t \in \mathbb{N}}$ and, possibly, the internal randomization used to choose the trading prices $(P_t)_{t \in \mathbb{N}}$.

### 1.2. Challenges and contributions

Under the assumption that the traders' valuations are unknown linear functions of $d$-dimensional contexts perturbed by zero-mean noise with time-variable densities bounded by

---

[2]We remark that in most markets, traders are allowed to sell assets they do not currently own (*short-selling*; see, e.g., the classic Black & Scholes 1973). For this reason, we do not need to assume that traders entering the market own a unit of the asset.

[3]We remark that we are not assuming that the two processes $(\xi_t)_{t \in \mathbb{N}}$ and $(\zeta)_{t \in \mathbb{N}}$ are i.i.d., and in fact the distributions of these random variables may change adversarially over time.

[4]Note that, mathematically, being competitive against a sequence of deterministic contexts is essentially the most general achievement that can be obtained, covering, in particular, the widespread and interpretable setting of i.i.d. random contexts as a special case.

[5]Although the market value is a linear function of the contexts, we remark that it is not necessarily true that the price $p^\star$ that maximizes the total expected gain from trade is also a linear function of the contexts, let alone that this function is the linear function that maps contexts to market values! One of the contributions of this work is to prove that this is true under the assumption that the noise distributions admit bounded densities (Lemma 2.1), but it becomes false when this assumption is lifted (Example 5.1).

| | Bounded density | General |
|---|---|---|
| **2-bit feedback** | $\sqrt{LdT}$ | $T$ |
| **Full feedback** | $Ld\ln T$ | $T$ |

*Table 1.* Summary of our results.

some $L$, we make the following contributions (see Table 1 for a summary).

1. We prove a key structural result (Lemma 2.1) with two crucial consequences. First, Lemma 2.1 shows that posting the (unknown) market value as the trading price would maximize the expected gain from trade.[6] Second, it proves that the loss paid by posting a suboptimal price is at most quadratic in the distance from the market value.

2. In our problem, the prices we post directly affect the two bits of information we retrieve (2-bit feedback). We note that this information is so scarce that it is not even enough to reconstruct *bandit* feedback. We solve this challenging exploration-exploitation dilemma by proposing an algorithm (Algorithm 1) that decides to either explore or exploit adaptively, based on the amount of contextual information gathered so far, and prove its optimality by showing a $\sqrt{LdT\ln T}$ regret upper bound (Theorem 3.1) and a matching (up to a $\sqrt{\ln T}$) $\sqrt{LdT}$ lower bound (Theorem 3.2).

3. To compare and contrast the impact of our realistic 2-bit feedback in online contextual brokerage, we investigate the rates that could be achieved if the traders' valuations were revealed at the end of any interaction (full feedback); for this problem, we prove that the optimal achievable rate is exponentially faster: of order $Ld\ln T$, by proving matching regret upper and lower bounds (Theorems 4.1 and 4.2).

4. Finally, we investigate the necessity of the bounded density assumption: by lifting this assumption, we show that the problem becomes unlearnable (Theorem 5.2), even under full feedback.

We stress that, in all our results, the dependence on *all* relevant parameters is tight. In contrast, as we discuss in Section 1.3, most related works on bilateral trade obtain (at best) a matching dependence in the time horizon only.

---

[6]This implies, in particular, that in our contextual setting where market prices are functions of contexts, the benchmark in the regret definition is the total expected reward of the best *arbitrary sequence* of prices, a benchmark that would be unattainable in similar problems (like standard adversarial bandits). This is one of the many differences between contextual and non-contextual settings.

## 1.3. Related Works

Our work extends the recent research on online learning algorithms for bilateral trade, in particular, Cesa-Bianchi et al. (2021); Azar et al. (2022); Cesa-Bianchi et al. (2024b; 2023; 2024c); Bernasconi et al. (2024), for non-contextual problems where sellers and buyers have definite roles. The stochastic case, where sellers' and buyers' valuations are i.i.d. across time, is studied in Cesa-Bianchi et al. (2021; 2024b). They obtain a $\sqrt{T}$ regret rate in the full-feedback setting. For the two-bit feedback case, they prove a linear worst-case lower bound, but it turns out that a tight regret rate of $T^{2/3}$ is possible, by assuming independence and uniformly bounded density for the sellers' and buyers' valuations. The adversarial setting is the topic of several works. In the worst-case, it is unlearnable as shown in Cesa-Bianchi et al. (2021; 2024b). Nevertheless, more favorable results exist under various relaxations. Cesa-Bianchi et al. (2023; 2024c) consider the adversarial case where the adversary is forced to be *smooth*, i.e., the sellers' and buyers' valuation distributions are allowed to change adversarially over time, but these distributions admit uniformly bounded densities. In the full-feedback case, they prove a tight $\sqrt{T}$ regret rate. In the two-bit feedback case, while the problem is still unlearnable, they allow the learner to use weakly budget-balanced mechanisms, yielding a surprisingly sharp $T^{3/4}$ regret rate. We remark that in all the two-bit feedback upper bounds requiring a bounded density assumption discussed above, there are no corresponding lower bounds with a sharp dependence on the density bound. Azar et al. (2022) consider the $\alpha$-regret objective, weaker than the regret. In the full-feedback case, they prove a tight 2-regret rate of $\sqrt{T}$. In the two-bit feedback case, while learning is impossible in general, they allow the learner to use weakly budget-balanced mechanisms, enabling to recover a 2-regret of order $T^{3/4}$. No matching lower bound is provided. Bernasconi et al. (2024) further relax the notion of weak budget-balance by proposing the notion of global budget-balance. Under global budget-balance, they provide a tight regret rate of $\sqrt{T}$ in the full-feedback case, and a regret rate of $T^{3/4}$ in the two-bit feedback case, without a matching lower bound.

Gaucher et al. (2025) investigated a noisy linear contextual version of the bilateral trade problem, where the authors obtain a tight regret bound (up to logarithmic factors) in the time horizon of order $T^{2/3}$ under 2-bit feedback, with mismatching dependence on the dimension and on the bounded density parameter in the lower bound. Even though their algorithm can be adapted to our setting (via the reduction that sets the seller's valuation as the minimum and the buyer's as the maximum of the traders' valuations), their regret guarantees (that would anyway be worse than our $\sqrt{T}$) are lost because they require that, conditioned to the context, the seller is independent of the buyer, which is not the case

in the reduction because, in general, the minimum of two random variables is not independent of the maximum of the same two random variables.

The brokerage problem in online learning has been introduced by Bolić et al. (2024) in a simpler i.i.d. and non-contextual setting. There, the authors study the non-contextual version of our trading problem with flexible sellers' and buyers' roles, with the further assumption that the sellers' and buyers' valuations form an i.i.d. sequence. Under the $M$-bounded density assumption, they obtain tight $M \ln T$ and $\sqrt{MT}$ regret rates in the full-feedback and two-bit feedback settings, respectively. If the bounded density assumption is removed, they show that the learning rate degrades to $\sqrt{T}$ in the full-feedback case and the problem turns out to be unlearnable in the two-bit feedback case. We remark that, interestingly, under the bounded density assumption, we are able to achieve the same regret rates in the contextual version of this problem without requiring that traders share the same valuation distribution, while, without the bounded density assumption, the contextual problem is unlearnable even under full feedback.

The non-contextual brokerage problem has also been recently studied with a different reward function aiming at maximizing the total volume of trades (Cesari & Colomboni, 2025).

Our linear assumption appears commonly in the literature on digital markets, particularly in problems like pricing and auctions. In Cohen et al. (2016; 2020), the authors first address a deterministic setting, then a noisy one with *known* noise distribution where they obtain a regret rate of order $T^{2/3}$ without presenting a lower bound. The deterministic case has also been investigated in Lobel et al. (2017; 2018); Leme & Schneider (2018; 2022); Liu et al. (2021).

The case of noisy linear functions has been studied in Xu & Wang (2021); Badanidiyuru et al. (2023); Fan et al. (2024); Luo et al. (2024); Chen & Gallego (2021); Javanmard & Nazerzadeh (2019); Bu et al. (2022); Shah et al. (2019) with guarantees limited to parametric or semi-parametric noise settings, while the recent work of Tullii et al. (2024) has given the first near-tight $T^{2/3}$ analysis of the non-parametric noise case.

The only work addressing contextual brokerage is Bachoc et al. (2025), which considers a non-parametric variant of our setting and derives tight (albeit considerably slower) regret bounds. Notably, their Theorem 5, combined with our Theorem 3.1 and Theorem 4.1, yields 2-regret guarantees against oracle policies that know the traders' valuations before setting prices.

Another rich related field explored in its many variants (Hanna et al., 2023; Slivkins et al., 2023; Leme et al., 2022; Foster et al., 2021; 2019; Zhou et al., 2019; Kirschner & Krause, 2019; Metevier et al., 2019; Foster & Krishnamurthy, 2018; Kannan et al., 2018; Oh & Iyengar, 2019; Hu et al., 2020; Neu & Olkhovskaya, 2020; Wei et al., 2020; Krishnamurthy et al., 2020; Luo et al., 2018; Krishnamurthy et al., 2021) is contextual linear bandits. In its standard form, at the beginning of each round, an action set is revealed to the learner, and the assumption is that the reward (which equals the feedback) is a linear function of the action selected from the action set. Instead, in our setting, the market price is a linear function of the context, while the rewards are linked to the price the learner posts by the non-linear gain from trade function. Moreover, in contrast to contextual bandits, in our 2-bit feedback model, the feedback differs from and is not sufficient to compute the reward of the action the learner selects at every round. For these reasons, existing theoretical results from contextual linear bandits do not directly apply to our problem. Nevertheless, note that techniques from contextual linear bandits are relevant to our problem, for instance, the use of the elliptical potential lemma (proof of Theorem 3.1).

Previously to online learning contributions, a fair amount of literature addressed game-theoretic and best-approximation aspects of bilateral trade. We refer in particular to the landmark work of Myerson and Satterthwaite (Myerson & Satterthwaite, 1983), as well as Colini-Baldeschi et al. (2016; 2017); Blumrosen & Mizrahi (2016); Brustle et al. (2017); Colini-Baldeschi et al. (2020); Babaioff et al. (2020); Dütting et al. (2021); Deng et al. (2022); Kang et al. (2022); Archbold et al. (2023). We also refer to Cesa-Bianchi et al. (2024b) for an analysis of the references above.

Finally, we point out that Amin et al. (2013); Golrezaei et al. (2019) address pricing problems related to brokerage and bilateral trade, and account for strategic aspects in this context.

## 2. Structural and Technical Results

We begin by presenting a structural result whose economic interpretation is as follows: even if the broker does not know the traders' valuation distributions, if these valuations can be modeled as zero-mean noisy perturbations with bounded densities of some market value, then the best price to post to maximize the expected gain from trade is precisely the (unknown and time-varying) market value, and the cost of posting a suboptimal price is at most quadratic in the distance from the market value.

In particular, this generalizes a similar result appearing in Bolić et al. (2024), which can be applied only under the further assumption that, at any time step, the traders' valuations have the exact same distribution. We argue that this assumption might be overly strong and not capture real-life behavior. This is because traders might have private

preferences or contingent needs that are not known by the broker; they could be more or less volatile, have differently skewed opinions, have valuations with arbitrarily different tail behavior, etc. Instead, we merely assume that, at any time step, traders' valuations are, on average, equal to the market price (which is how market prices are essentially determined in real life) but allow for arbitrarily different (hidden and time-varying) distributions. This relaxation of the assumption comes at the expense of a more subtle proof. The mathematical reason for the added difficulty is that, under the same-distribution assumption, many of the terms appearing in the proof simplify due to symmetries, while, in our case, a different approach is needed to recover all the properties needed for our result (which we are able to obtain without introducing any new assumptions).

This structural result is the key to unraveling the intricacies of the noisy contextual setting, and it is what ultimately allows us to obtain tight regret guarantees in all settings, distinguishing ours from similar contextual bilateral trade and pricing works.

**Lemma 2.1.** *Suppose that $V$ and $W$ are two $[0,1]$-valued independent random variables, with possibly different densities bounded by some constant $L \geq 1$, and such that $\mathbb{E}[V] = \mathbb{E}[W] =: m$. Then, for each $p \in [0,1]$, it holds that*

$$0 \leq \mathbb{E}\big[g(m, V, W) - g(p, V, W)\big] \leq L\,|m - p|^2 \ .$$

Due to space constraints, we defer the technical proof of this lemma to Appendix A.1.

As an immediate corollary of Lemma 2.1, we obtain the following important result that upper bounds the regret in terms of the sum of the squared distances between the prices the algorithm posts and the actual market values.

**Corollary 2.2.** *Consider the contextual brokerage problem introduced in Section 1.1. If the valuations admit densities bounded by a constant $L \geq 1$, then, for any time horizon $T \in \mathbb{N}$, we have*

$$R_T = \mathbb{E}\left[\sum_{t=1}^{T}\big(\mathrm{GFT}_t(c_t^\top \phi) - \mathrm{GFT}_t(P_t)\big)\right]$$
$$\leq \sum_{t=1}^{T} 1 \wedge \Big(L\mathbb{E}\big[|P_t - c_t^\top \phi|^2\big]\Big) \ .$$

*Proof.* Given that for each $t \in \mathbb{N}$ and each $p \in [0,1]$ it holds that $\mathrm{GFT}_t(p) \in [0,1]$, we have $\sup_{p \in [0,1]} \mathbb{E}\big[\mathrm{GFT}_t(p) - \mathrm{GFT}_t(P_t)\big] \leq 1$, and hence, recalling that $m_t = c_t^\top \phi$ and

that $\mathbb{E}[V_t] = m_t = \mathbb{E}[W_t]$, we also have, for each $T \in \mathbb{N}$,

$$R_T = \sup_{p^\star} \sum_{t=1}^{T} 1 \wedge \Big(\mathbb{E}\big[g(p^\star(c_t), V_t, W_t)\big] - \mathbb{E}\big[g(P_t, V_t, W_t)\big]\Big)$$
$$\stackrel{(\circ)}{=} \sum_{t=1}^{T} 1 \wedge \Big(\mathbb{E}\big[g(c_t^\top \phi, V_t, W_t)\big] - \mathbb{E}\big[g(P_t, V_t, W_t)\big]\Big)$$
$$\stackrel{(*)}{=} \sum_{t=1}^{T} 1 \wedge \mathbb{E}\left[\big[\mathbb{E}\big[g(c_t^\top \phi, V_t, W_t) - g(p, V_t, W_t)\big]\big]_{p=P_t}\right]$$
$$\stackrel{(\circ)}{\leq} \sum_{t=1}^{T} 1 \wedge \Big(L\mathbb{E}\big[|P_t - c_t^\top \phi|^2\big]\Big) \ ,$$

where the supremum in the first equality is over all functions $p^\star \colon [0,1]^d \to [0,1]$, $(\circ)$ is a directed consequence of Lemma 2.1, and $(*)$ follows from the Freezing Lemma (Cesari & Colomboni, 2021, Lemma 8). $\qquad\square$

We conclude this section by presenting the following technical lemma, which will be used in the analyses of our Algorithms 1 and 2 to control the behavior of the estimators we employ. Its proof is deferred to Appendix A.2.

We write, for any $l \in \mathbb{N}$, $\mathbf{1}_l$ for the $l$-dimensional identity matrix. Also, for any positive definite matrix $A \in \mathbb{R}^{l \times l}$, we define $\|\cdot\|_A \colon \mathbb{R}^l \to [0, \infty)$, $v \mapsto \sqrt{v^\top A v}$.

**Lemma 2.3.** *Let $s, l \in \mathbb{N}$. Let $Z_1, \ldots, Z_s$ be an independent sequence of $[0,1]$-valued random variables. Let $a_1, \ldots, a_s \in [0,1]^l$. Let $\psi \in [0,1]^l$. Suppose that, for each $r \in [s]$ it holds that $\mathbb{E}[Z_r] = a_r^\top \psi$. Define $f_s \coloneqq [a_1 \mid \cdots \mid a_s]$. Define $H_s \coloneqq [Z_1 \mid \cdots \mid Z_s]$. Define $\hat{\psi}_s \coloneqq (f_s f_s^\top + l^{-1}\mathbf{1}_l)^{-1} f_s H_s^\top$. Then, if $a \in [0,1]^l$, we have that*

$$\mathbb{E}\big[|a^\top \hat{\psi}_s - a^\top \psi|^2\big] \leq \big\|\sqrt{2}a\big\|^2_{\left(\sum_{r=1}^{s} a_r a_r^\top + l^{-1}\mathbf{1}_l\right)^{-1}} \ .$$

## 3. Learning in Contextual Brokerage

In this section, we introduce an algorithm (Algorithm 1) for the contextual brokerage problem for which we prove regret guarantees of order $\widetilde{\mathcal{O}}\big(\sqrt{LdT}\big)$. The key feature of the algorithm's design is a deterministic rule that decides to either explore or exploit based on the amount of information gathered along the various context directions (see the definition of $b_t$ on Line 4). When the algorithm explores, it posts a price drawn uniformly in $[0,1]$ to obtain an unbiased estimate of the current market value. When it exploits, it posts the scalar product of the context and the current estimate of the unknown weight vector $\phi$ built using the information retrieved during exploration rounds.

**Theorem 3.1.** *If the learner runs Algorithm 1 and the traders' valuations admit a density bounded by $L \geq 1$, then, for any time horizon $T$ such that $LT \geq 2d \ln\big(1 + 2d(T-1)\big)$, it holds that $R_T \leq 1 + 6\sqrt{LdT \ln T}$.*

**Algorithm 1**

1: Post $P_1$ uniformly at random in $[0, 1]$, and observe $D_1 \coloneqq \mathbb{I}\{P_1 \le V_1\}$
2: Let $b_1 \coloneqq 1$, let $x_1 \coloneqq [c_1]$, let $Y_1 \coloneqq [D_1]$ and compute $\hat{\phi}_1 \coloneqq (x_1 x_1^\top + d^{-1} \mathbf{1}_d)^{-1} x_1 Y_1^\top$
3: **for** time $t = 2, 3, \dots$ **do**
4:   Observe context $c_t$ and define $b_t \coloneqq \mathbb{I}\left\{ \left\| \sqrt{2} c_t \right\|^2_{(x_{t-1} x_{t-1}^\top + d^{-1} \mathbf{1}_d)^{-1}} > \sqrt{\frac{2d \ln(1 + 2d(T-1))}{LT}} \right\}$
5:   **if** $b_t = 1$ **then**
6:     Post $P_t$ uniformly at random in $[0, 1]$, and observe $D_t \coloneqq \mathbb{I}\{P_t \le V_t\}$
7:     Let $x_t \coloneqq [x_{t-1} \mid c_t]$, let $Y_t \coloneqq [Y_{t-1} \mid D_t]$ and compute $\hat{\phi}_t \coloneqq (x_t x_t^\top + \mathbf{1}_d)^{-1} x_t Y_t^\top$
8:   **else**
9:     post $P_t \coloneqq c_t^\top \hat{\phi}_{t-1}$ and let $x_t \coloneqq x_{t-1}$, $Y_t \coloneqq Y_{t-1}$, and $\hat{\phi}_t \coloneqq \hat{\phi}_{t-1}$
10:   **end if**
11: **end for**

*Proof.* Without loss of generality we assume that $T \ge 2$. Note that for any $t \in \mathbb{N}$, if $b_t = 1$, then

$$
\mathbb{E}[D_t] = \mathbb{P}[P_t \le V_t]
$$
$$
= \int_0^1 \mathbb{P}[u \le V_t] \, \mathrm{d}u = \mathbb{E}[V_t] = \mathbb{E}[c_t^\top \phi + \xi_t] = c_t^\top \phi . \quad (1)
$$

Now, fix $t \ge 2$. Let $s \coloneqq \sum_{i=1}^{t-1} b_i$ be the total number of exploration steps done before time step $t$. Define recursively time steps $\tau(1), \dots, \tau(s)$ as follows: let $\tau(1) = 1$ and, for all $n \in [s-1]$, define $\tau(n+1) \coloneqq \min\{i \in [t-1] \mid i \ge \tau(n) + 1, b_i = 1\}$. Now, for each $n \in [s]$, define $Z_n \coloneqq D_{\tau(n)}$. Notice that $Z_1, \dots, Z_s$ are well defined because for each $n \in [s]$ we have that $b_{\tau(n)} = 1$. Define $l \coloneqq d$. For each $n \in [s]$, define $a_n \coloneqq c_{\tau(n)}$. Let $\psi \coloneqq \phi$ and $a \coloneqq c_t$. Notice that, by Equation (1), if $j \in [s]$ is odd, then $\mathbb{E}[Z_j] = \mathbb{E}[D_{\tau(j)}] = c_{\tau(j)}^\top \phi = a_j^\top \psi$. Then, with the same notation of Lemma 2.3, we can apply Lemma 2.3 to obtain

$$
\mathbb{E}[|c_t^\top \hat{\phi}_{t-1} - c_t^\top \phi|^2] = \mathbb{E}[|a^\top \hat{\psi}_s - a^\top \psi|^2]
$$
$$
\le \left\| \sqrt{2} a \right\|^2_{\left(\sum_{j=1}^s a_j a_j^\top + l^{-1} \mathbf{1}_l\right)^{-1}}
$$
$$
= \left\| \sqrt{2} c_t \right\|^2_{\left(\sum_{n=1}^s c_{\tau(n)} c_{\tau(n)}^\top + d^{-1} \mathbf{1}_d\right)^{-1}}
$$
$$
= \left\| \sqrt{2} c_t \right\|^2_{\left(\sum_{i=1}^{t-1} b_i c_i c_i^\top + d^{-1} \mathbf{1}_d\right)^{-1}} = \left\| \sqrt{2} c_t \right\|^2_{\left(x_{t-1} x_{t-1}^\top + d^{-1} \mathbf{1}_d\right)^{-1}}
$$

where the last step follows by definition of $x_{t-1}$.

Being $t$ arbitrarily chosen, we have that for each $t \in [T]$ such that $t \ge 2$,

$$
\mathbb{E}[|c_t^\top \hat{\phi}_{t-1} - c_t^\top \phi|^2] \le \left\| \sqrt{2} c_t \right\|^2_{\left(x_{t-1} x_{t-1}^\top + d^{-1} \mathbf{1}_d\right)^{-1}} .
$$

Hence, leveraging Corollary 2.2 and the previous inequality, for any $T \in \mathbb{N}$, we have that

$$
R_T \le \sum_{t=1}^T 1 \wedge \left( L \mathbb{E}[|P_t - c_t^\top \phi|^2] \right)
$$
$$
\le \sum_{t=2}^T (1 - b_t) L \mathbb{E}\left[ |c_t^\top \hat{\phi}_{t-1} - c_t^\top \phi|^2 \right] + \sum_{t=1}^T b_t
$$
$$
\le L \sum_{t=2}^T (1 - b_t) \left\| \sqrt{2} c_t \right\|^2_{\left(x_{t-1} x_{t-1}^\top + d^{-1} \mathbf{1}_d\right)^{-1}} + \sum_{t=1}^T b_t
$$
$$
\le \sqrt{2 L d T \ln\left(1 + 2d(T-1)\right)} + \sum_{t=1}^T b_t ,
$$

where in the last step we used the definition of the $b_1, \dots, b_T$. Now, given that $LT / \left(2d \ln\left(1 + 2d(T-1)\right)\right) \ge 1$, using the convention $0/0 = 0$,

$$
\sum_{t=2}^T b_t = \sum_{t=2}^T \frac{b_t \left\| \sqrt{2} c_t \right\|^2_{(x_{t-1} x_{t-1}^\top + d^{-1} \mathbf{1}_d)^{-1}}}{\left\| \sqrt{2} c_t \right\|^2_{(x_{t-1} x_{t-1}^\top + d^{-1} \mathbf{1}_d)^{-1}}}
$$
$$
\le \sqrt{\frac{LT}{2d \ln(1 + 2d(T-1))}}
$$
$$
\cdot \sum_{t=2}^T 1 \wedge b_t \left\| \sqrt{2} c_t \right\|^2_{\left(\sum_{s=1}^{t-1} b_s c_s c_s^\top + d^{-1} \mathbf{1}_d\right)^{-1}}
$$
$$
\le \sqrt{\frac{2LT}{d \ln\left(1 + 2d(T-1)\right)}}
$$
$$
\cdot \sum_{t=1}^{T-1} 1 \wedge \left\| b_{t+1} c_{t+1} \right\|^2_{\left(\sum_{s=1}^t (b_s c_s)(b_s c_s)^\top + d^{-1} \mathbf{1}_d\right)^{-1}}
$$
$$
=: (\star).
$$

Using the elliptical potential lemma (Lattimore & Szepesvári, 2020, Lemma 19.4), we obtain

$$
\sum_{t=1}^T b_t \le 1 + (\star)
$$
$$
\le 1 + \sqrt{2LT / \left(d \ln\left(1 + 2d(T-1)\right)\right) \cdot 2d \ln\left(1 + 2d(T-1)\right)}
$$
$$
= 1 + 2\sqrt{2 L d T \ln\left(1 + 2d(T-1)\right)} .
$$

Hence, if $d < T/2$, this implies that $R_T \le 1 + 3\sqrt{2 L d T \ln\left(1 + 2d(T-1)\right)} \le 1 + 6\sqrt{L d T \ln T}$. On the other hand, if $d \ge T/2$, then, since $L \ge 1$, we obtain, again, $R_T \le T \le 1 + 6\sqrt{L d T \ln T}$. $\qquad \square$

We conclude this section by stating a matching (up to logarithmic terms) worst-case $\Omega\left(\sqrt{L d T}\right)$ regret lower bound for any algorithm, proving the optimality of Algorithm 1.

At a high level, the proof of this result is based on first building a sequence of contexts defined as a common element of the canonical basis of $\mathbb{R}^d$ during each one of $d$ blocks of

$T/d$ consecutive time steps. Then, in each block, we use a non-contextual lower bound construction leading to a regret of at least $\sqrt{LT/d}$ for each block, and conclude the proof by summing over blocks. For more details on the proof of this result, see Appendix C.2.

**Theorem 3.2.** *There exist two numerical constants $a, b > 0$ such that, for any $L \geq 2$ and any time horizon $T \geq \max(4, adL^3, 2d)$, there exists a sequence of contexts $c_1, \ldots, c_T \in [0,1]^d$ such that, for any algorithm $\alpha$ for the contextual brokerage problem, there exists a vector $\phi \in [0,1]^d$ and two zero-mean independent sequences $(\xi_t)_{t\in[T]}$ and $(\zeta_t)_{t\in[T]}$ independent of each other such that, if we define $V_t := c_t^\top \phi + \xi_t$ and $W_t := c_t^\top \phi + \zeta_t$, then for each $t \in [T]$ it holds that $c_t^\top \phi \in [0,1]$, $V_t$ and $W_t$ are $[0,1]$-valued random variables with density bounded by $L$, and the regret of $\alpha$ on the sequence of traders' valuations $V_1, W_1, \ldots, V_T, W_T$ satisfies $R_T \geq b\sqrt{LdT}$.*

We remark that the previous lower bound holds even for algorithms that have prior knowledge of the sequence of contexts $c_1, c_2, \ldots$ and that Theorem 3.1 shows that Algorithm 1 matches the optimal $\sqrt{LdT}$ rate (up to a $\sqrt{\ln T}$ factor) even without this *a-priori* knowledge.

## 4. Full Feedback

In this section, we discuss a "full feedback" version of the contextual brokerage problem to understand how the limited feedback the broker has normally access to impacts the regret. In this version, the valuations $V_t$ and $W_t$ are revealed at the end of each time step $t$.

For this problem, we modify Algorithm 1 in two ways to leverage the higher-quality feedback. First, the new algorithm never explores (it does not need to), i.e., $b_t := 0$ for all $t$. Second, the algorithm uses different (and better) unbiased estimators of $m_t$ in the columns of $Y_t$: the valuations $V_t$ and $W_t$. The resulting algorithm is Algorithm 2, for which we prove an optimal *logarithmic* worst-case regret: an exponential improvement with respect to what is achievable under the classic 2-bit feedback.

---

**Algorithm 2**

1: Observe context $c_1$, post $P_1 := 1/2$, and receive feedback $V_1, W_1$
2: Let $x_1 := [c_1 \mid c_1]$, let $Y_1 := [V_1 \mid W_1]$, and compute $\hat{\phi}_1 := (x_1 x_1^\top + d^{-1}\mathbf{1}_d)^{-1} x_1 Y_1^\top$
3: **for** time $t = 2, 3, \ldots$ **do**
4:     Observe context $c_t$, post $P_t := c_t^\top \hat{\phi}_{t-1}$, and receive feedback $V_t, W_t$
5:     Let $x_t := [x_{t-1} \mid c_t \mid c_t]$, $Y_t := [Y_{t-1} \mid V_t \mid W_t]$, and compute $\hat{\phi}_t := (x_t x_t^\top + d^{-1}\mathbf{1}_d)^{-1} x_t Y_t^\top$
6: **end for**

---

**Theorem 4.1.** *Consider the full-feedback version of the contextual brokerage problem. If the learner runs Algorithm 2 and the traders' valuations admit a density bounded by $L \geq 1$, then, for any time horizon $T \in \mathbb{N}$, it holds that $R_T \leq 1 + 4Ld\ln T$.*

Due to space constraints, we defer the proof of this result to Appendix B.1.

We conclude this section by stating a matching worst-case $\Omega(Ld\ln T)$ regret lower bound for any algorithm in the full-feedback case, proving the optimality of Algorithm 2.

The result is proved similarly to that of Theorem 3.2: first, in each one of $d$ blocks of $T/d$ consecutive time steps, we play a fixed context defined as a common element of the canonical basis of $\mathbb{R}^d$. Then, in each block, we use a non-contextual lower bound construction leading to a regret of at least $L\ln(T/d)$ for the block, and conclude the proof by summing over blocks. For more details on the proof of this result, see Appendix C.1.

**Theorem 4.2.** *There exist two numerical constants $a, b > 0$ such that, for any $L \geq 2$ and any time horizon $T \geq \max(4, adL^5, 2d)$, there exists a sequence of contexts $c_1, \ldots, c_T \in [0,1]^d$ such that, for any algorithm $\alpha$ for the full-feedback version of the contextual brokerage problem, there exists a vector $\phi \in [0,1]^d$ and two zero-mean independent sequences $(\xi_t)_{t\in[T]}$ and $(\zeta_t)_{t\in[T]}$ independent of each other, such that if we define $V_t := c_t^\top \phi + \xi_t$ and $W_t := c_t^\top \phi + \zeta_t$, then for each $t \in [T]$ it holds that $c_t^\top \phi \in [0,1]$, $V_t$ and $W_t$ are $[0,1]$-valued random variables with density bounded by $L$, and the regret of $\alpha$ on the sequence of traders' valuations $V_1, W_1, \ldots, V_T, W_T$ satisfies $R_T \geq bLd\ln T$.*

We remark that the previous lower bound holds even for algorithms that have prior knowledge of the sequence of contexts $c_1, c_2, \ldots$ and that Theorem 4.1 shows that Algorithm 2 matches the optimal $Ld\ln T$ rate even without this *a-priori* knowledge.

## 5. Beyond Bounded Densities

In this final section, we investigate the general case where the traders' valuations are not assumed to have a bounded density. We begin with the following (perhaps counterintuitive) counterexample showing that, in general, posting the market value can be highly suboptimal if the goal is to maximize the gain from trade.

*Example* 5.1. Let $V$ and $W$ be two independent uniform random variables on $\{0, \frac{1}{5}, 1\}$ and $m := \mathbb{E}[V] = \mathbb{E}[W] = 2/5$. Then $\operatorname{argmax}_{p\in[0,1]} \mathbb{E}[g(p, V, W)] = 1/5 \neq m$, and $\mathbb{E}[g(1/5, V, W)] - \mathbb{E}[g(2/5, V, W)] = 2 \cdot \left(\frac{1}{5} - 0\right) \cdot \frac{1}{9} > 0$.

The phenomenon illustrated by the previous counterexample is the key to proving our final result: the unlearnability, in general, of the brokerage problem (even when full feedback

is available). Specifically, we exploit the fact that, in general, the optimal price at time $t$ depends not only on the market value $m_t = c_t^\top \phi$ but also on properties of the *adversarial and time-varying* distributions of the perturbations $\xi_t$ and $\zeta_t$, which make it impossible to compete against the benchmark in our regret definition.

**Theorem 5.2.** *There exists a sequence of contexts $c_1, c_2, \dots \in [0,1]^d$ and a vector $\phi \in [0,1]^d$, such that for any algorithm $\alpha$ for the full-feedback version of the contextual brokerage problem, there exists an independent sequence of zero mean random variables $\xi_1, \zeta_1, \xi_2, \zeta_2, \dots$, such that if the valuations of the traders at time $t$ are $V_t = c_t^\top \phi + \xi_t$ and $W_t = c_t^\top \phi + \zeta_t$, then $c_t^\top \phi \in [0,1]$, $V_t, W_t$ are $[0,1]$-valued random variables, and the regret of $\alpha$ on the sequence of traders' valuations $V_1, W_1, \dots, V_T, W_T$ satisfies $R_T \geq \frac{1}{32}T$.*

We remark that the previous unlearnability result holds even for algorithms that have prior knowledge of the sequence of contexts $c_1, c_2, \dots$ and, strikingly, of the vector $\phi$, that is, even for algorithms that know the entire sequence $m_1, m_2, \dots$ of market prices in advance!

*Proof.* Assume that $d \geq 2$ (for the case $d = 1$, the following proof can be adapted straightforwardly by defining $\phi = 1$ and $c_t = 1/2 + \varepsilon_t$, where $\varepsilon_t$ is an arbitrary small sequence of biases). Let $(a_t)_{t \in \mathbb{N}}$ be a sequence of distinct elements in $[0,1]$ and, for all $t \in \mathbb{N}$, let $c_t := (a_t, 1 - a_t, 0, 0, \dots, 0)$. Notice that $(c_t)_{t \in \mathbb{N}}$ is a sequence of distinct elements in $[0,1]^d$. Define $\phi := (1/2, 1/2, 0, 0, \dots, 0)$. Notice that for each $t \in \mathbb{N}$ it holds that $c_t^\top \phi = 1/2$. Let $\varepsilon \in (0, 1/16)$. For any $\theta \in \{0,1\}$, consider the following probability distribution

$$\mu_\theta := \left(\tfrac{1}{4} + (1 - 2\theta)\varepsilon\right)\delta_{-\frac{1}{2}} + \tfrac{1}{2}\delta_{f(\theta)} + \left(\tfrac{1}{4} - (1 - 2\theta)\varepsilon\right)\delta_{\frac{1}{2}}$$

where $f(\theta) := 2(1 - \theta)\varepsilon - 2\theta\varepsilon$ and for any $a \in \mathbb{R}$, $\delta_a$ is the Dirac's delta probability distribution centered in $a$. Consider an independent family of random variables $(\xi_{t,\theta}, \zeta_{t,\theta})_{t \in \mathbb{N}, \theta \in \{0,1\}}$ such that for any $t \in \mathbb{N}$ and any $\theta \in \{0,1\}$, we have that both $\xi_{t,\theta}$ and $\zeta_{t,\theta}$ are random variables with common distribution $\mu_\theta$. Notice that for each $t \in \mathbb{N}$ and each $\theta \in \{0,1\}$ we have that $\mathbb{E}[\xi_{t,\theta}] = 0 = \mathbb{E}[\zeta_{t,\theta}]$. Define, for each $t \in \mathbb{N}$ and each $\theta \in \{0,1\}$, the random variables $V_{t,\theta} := c_t^\top \phi + \xi_{t,\theta}$ and $W_{t,\theta} := c_t^\top \phi + \zeta_{t,\theta}$. Notice that these are $[0,1]$-valued random variables and that $(V_{t,\theta}, W_{t,\theta})_{t \in \mathbb{N}, \theta \in \{0,1\}}$ is an independent family. Now, for each $\theta \in \{0,1\}$ and each $t \in \mathbb{N}$, let $p \mapsto G_t^\theta(p) := g(p, V_{t,\theta}, W_{t,\theta})$ and

$$p^\#(\theta) \in \operatorname{argmax}_{p \in [0,1]} \mathbb{E}[G_t^\theta(p)],$$

which does exist because the function $[0,1] \to [0,1], p \mapsto \mathbb{E}[G_t^\theta(p)]$ is upper semicontinuous (this can be proved, e.g., as in Cesa-Bianchi et al. 2024b, Appendix B) and defined on a compact set. Furthermore, note that the previous definition is independent of $t$ because, for any $\theta \in \{0,1\}$, the

pairs $(V_{t_1,\theta}, W_{t_1,\theta})$ and $(V_{t_2,\theta}, W_{t_2,\theta})$ share the same distribution for every $t_1, t_2 \in \mathbb{N}$. Fix a learning algorithm for the full-feedback contextual brokerage problem, fix a time horizon $T \in \mathbb{N}$, and notice that since the contexts $c_1, c_2, \dots$ are all distinct, it follows that

$$\max_{\theta_1, \dots, \theta_T \in \{0,1\}^T} \sup_{p^\star : [0,1]^d \to [0,1]} \mathbb{E}\left[\sum_{t=1}^T \left(G_t^{\theta_t}\left(p^\star(c_t)\right) - G_t^{\theta_t}(P_t)\right)\right]$$

$$= \max_{\theta_1, \dots, \theta_T \in \{0,1\}^T} \sum_{t=1}^T \left(\sup_{p \in [0,1]} \mathbb{E}[G_t^{\theta_t}(p)] - \mathbb{E}[G_t^{\theta_t}(P_t)]\right)$$

$$= \max_{\theta_1, \dots, \theta_T \in \{0,1\}^T} \sum_{t=1}^T \mathbb{E}\left[G_t^{\theta_t}\left(p^\#(\theta_t)\right) - G_t^{\theta_t}(P_t)\right] =: (\#) .$$

Now, consider an i.i.d. family of Bernoulli random variables $(\Theta_t)_{t \in \mathbb{N}}$ with parameter $1/2$, independent of the whole family $(V_{t,\theta}, W_{t,\theta})_{t \in \mathbb{N}, \theta \in \{0,1\}}$. We have that

$$(\#) \geq \sum_{t=1}^T \left(\mathbb{E}\left[G_t^{\Theta_t}\left(p^\#(\Theta_t)\right)\right] - \mathbb{E}\left[G_t^{\Theta_t}(P_t)\right]\right) =: (\$).$$

Now, for each $t \in [T]$, we see that

$$\mathbb{E}\left[G_t^{\Theta_t}\left(p^\#(\Theta_t)\right)\right] = \mathbb{E}\left[\mathbb{E}\left[G_t^{\Theta_t}\left(p^\#(\Theta_t)\right) \mid \Theta_t\right]\right]$$

$$= \mathbb{E}\left[\max_{p \in [0,1]} \mathbb{E}\left[G_t^{\Theta_t}(p) \mid \Theta_t\right]\right]$$

and long but straightforward computations show that, for each $p \in [0,1]$, it holds that the conditional expectation $\mathbb{E}\left[G_t^{\Theta_t}(p) \mid \Theta_t\right]$ is equal to

$$\begin{cases} \frac{1}{4} + \varepsilon(1 - 2\Theta_t) & \text{if } 0 \leq p < \frac{1}{2} - 2\Theta_t\varepsilon + 2(1 - \Theta_t)\varepsilon , \\ \frac{3}{8} + 2\varepsilon^2 & \text{if } p = \frac{1}{2} - 2\Theta_t\varepsilon + 2(1 - \Theta_t)\varepsilon , \\ \frac{1}{4} - \varepsilon(1 - 2\Theta_t) & \text{if } \frac{1}{2} - 2\Theta_t\varepsilon + 2(1 - \Theta_t)\varepsilon < p \leq 1 , \end{cases}$$

from which it follows that

$$\max_{p \in [0,1]} \mathbb{E}\left[G_t^{\Theta_t}(p) \mid \Theta_t\right] = \frac{3}{8} + 2\varepsilon^2 .$$

On the other hand, for each $t \in [T]$, leveraging the freezing lemma (Cesari & Colomboni, 2021, Lemma 8), we have that

$$\mathbb{E}\left[G_t^{\Theta_t}(P_t)\right] = \mathbb{E}\left[\mathbb{E}\left[G_t^{\Theta_t}(P_t) \mid P_t\right]\right] = \mathbb{E}\left[\left[\mathbb{E}\left[G_t^{\Theta_t}(p)\right]\right]_{p = P_t}\right]$$

$$= \mathbb{E}\left[\left[\frac{1}{2}\mathbb{E}\left[G_t^{\Theta_t}(p) \mid \Theta_t = 0\right] + \frac{1}{2}\mathbb{E}\left[G_t^{\Theta_t}(p) \mid \Theta_t = 1\right]\right]_{p = P_t}\right]$$

and again, tedious but straightforward computations show

that, for each $p \in [0,1]$, it holds that

$$
\frac{1}{2}\mathbb{E}\big[G_t^{\Theta_t}(p) \mid \Theta_t = 0\big] + \frac{1}{2}\mathbb{E}\big[G_t^{\Theta_t}(p) \mid \Theta_t = 1\big]
$$
$$
= \frac{1}{4}\left(\mathbb{I}\left\{p < \tfrac{1}{2} - 2\varepsilon\right\} + \mathbb{I}\left\{\tfrac{1}{2} + 2\varepsilon < p\right\}\right)
$$
$$
+ \left(\tfrac{5}{16} + \tfrac{\varepsilon}{2} + \varepsilon^2\right)\left(\mathbb{I}\left\{p = \tfrac{1}{2} - 2\varepsilon\right\} + \mathbb{I}\left\{p = \tfrac{1}{2} + 2\varepsilon\right\}\right)
$$
$$
+ \left(\tfrac{1}{4} + \varepsilon\right)\mathbb{I}\left\{\tfrac{1}{2} - 2\varepsilon < p < \tfrac{1}{2} + 2\varepsilon\right\}
$$
$$
\le \tfrac{5}{16} + \tfrac{\varepsilon}{2} + \varepsilon^2 \,.
$$

We conclude that $(\$) \ge \frac{T}{16} + \left(\varepsilon^2 - \frac{\varepsilon}{2}\right)T \ge \frac{T}{32}$, from which it follows that there exists $\theta_1, \ldots, \theta_T \in \{0,1\}$ such that

$$
\sup_{p^\star : [0,1]^d \to [0,1]} \mathbb{E}\left[\sum_{t=1}^{T}\Big(G_t^{\theta_t}\big(p^\star(c_t)\big) - G_t^{\theta_t}(P_t)\Big)\right] \ge \frac{T}{32} \,. \quad \square
$$

## 6. Conclusions

Motivated by the real-life *desideratum* to exploit prior information on the traded assets, we investigated the noisy linear contextual online learning problem of brokerage between traders without predetermined seller/buyer roles. We provided a complete picture with tight regret bounds in all the proposed settings, achieving tightness (up to $\log$ terms) in all relevant parameters.

Our work stands on the classic interpretation of the market value of an asset as the average opinion of the market participants. An alternative perspective, which we leave open for future research, is when, instead, assets have an "inherent value", and traders' valuations are systematic biases or strategic deviations around this quantity. In this case, this inherent value would not be the average of the traders' valuations, and new techniques will be required to analyze this setting.

Finally, we highlight that there are many other online learning problems in digital markets whose contextual version is still open, such as market making (Cesa-Bianchi et al., 2025), first-price auctions with unknown costs (Cesa-Bianchi et al., 2024a), trading-volume maximization (Cesari & Colomboni, 2025), and optimal taxation (Cesa-Bianchi et al., 2025).

## Acknowledgements

The work of FB was supported by the Project GAP (ANR-21-CE40-0007) of the French National Research Agency (ANR) and by the Chair UQPhysAI of the Toulouse AN-ITI AI Cluster. TC gratefully acknowledges the support of the University of Ottawa through grant GR002837 (Start-Up Funds) and that of the Natural Sciences and Engineering Research Council of Canada (NSERC) through grants RGPIN-2023-03688 (Discovery Grants Program) and DGECR2023-00208 (Discovery Grants Program, DGECR - Discovery Launch Supplement). RC is partially supported by the MUR PRIN grant 2022EKNE5K (Learning in Markets and Society), the FAIR (Future Artificial Intelligence Research) project, funded by the NextGenerationEU program within the PNRR-PE-AI scheme, the EU Horizon CL4-2022-HUMAN-02 research and innovation action under grant agreement 101120237, project ELIAS (European Lighthouse of AI for Sustainability).

## Impact Statement

This paper presents work whose goal is to advance the field of Machine Learning. There are many potential societal consequences of our work, none of which we feel must be specifically highlighted here.

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

## A. Missing Proofs of Structural and Technical Results (Section 2)

In this section, we provide the missing proofs of our structural and technical results.

### A.1. Proof of Lemma 2.1

We denote by $F$ (resp., $G$) the cumulative distribution function of $V$ (resp., $W$). For each $p \in [0, 1]$, from the Decomposition Lemma in (Cesa-Bianchi et al., 2024b, Lemma 1), it holds that

$$\mathbb{E}\big[(W - V)\mathbb{I}\{V \le p \le W\}\big] = F(p) \int_p^1 \big(1 - G(\lambda)\big)\, d\lambda + \big(1 - G(p)\big) \int_0^p F(\lambda)\, d\lambda\,,$$

$$\mathbb{E}\big[(V - W)\mathbb{I}\{W \le p \le V\}\big] = G(p) \int_p^1 \big(1 - F(\lambda)\big)\, d\lambda + \big(1 - F(p)\big) \int_0^p G(\lambda)\, d\lambda\,.$$

Hence, for each $p \in [0, 1]$,

$$
\begin{aligned}
\mathbb{E}\big[(W - V)\mathbb{I}\{V \le p \le W\}\big] &= F(p) \int_p^1 \big(1 - G(\lambda)\big)\, d\lambda + \big(1 - G(p)\big) \int_0^p F(\lambda)\, d\lambda \\
&= F(p) \Big(m - \int_0^p \big(1 - G(\lambda)\big)\, d\lambda\Big) + \int_0^p F(\lambda)\, d\lambda - G(p) \int_0^p F(\lambda)\, d\lambda \\
&= \int_0^p F(\lambda)\, d\lambda + (m - p)F(p) - pG(p) + G(p) \int_0^p \big(1 - F(\lambda)\big)\, d\lambda + F(p) \int_0^p G(\lambda)\, d\lambda \\
&= \int_0^p (F + G)(\lambda)\, d\lambda + (m - p)(F + G)(p) - G(p)\Big(m - \int_0^p \big(1 - F(\lambda)\big)\, d\lambda\Big) + (F(p) - 1) \int_0^p G(\lambda)\, d\lambda \\
&= \int_0^p (F + G)(\lambda)\, d\lambda + (m - p)(F + G)(p) - \Big(G(p) \int_p^1 \big(1 - F(\lambda)\big)\, d\lambda + \big(1 - F(p)\big) \int_0^p G(\lambda)\, d\lambda\Big) \\
&= \int_0^p (F + G)(\lambda)\, d\lambda + (m - p)(F + G)(p) - \mathbb{E}\big[(V - W)\mathbb{I}\{W \le p \le V\}\big]\,.
\end{aligned}
$$

Rearranging, it follows that, for each $p \in [0, 1]$,

$$
\begin{aligned}
\mathbb{E}\big[g(p, V, W)\big] &= \mathbb{E}\big[(W - V)\mathbb{I}\{V \le p \le W\}\big] + \mathbb{E}\big[(V - W)\mathbb{I}\{W \le p \le V\}\big] \\
&= \int_0^p (F + G)(\lambda)\, d\lambda + (m - p)(F + G)(p)\,.
\end{aligned}
$$

Hence, for any $p \in [0, 1]$, it holds that

$$\mathbb{E}\big[g(m, V, W) - g(p, V, W)\big] = \int_p^m \big((F + G)(\lambda) - (F + G)(p)\big)\, d\lambda \ge 0\,.$$

Finally, since $F$ and $G$ are absolutely continuous with weak derivative bounded by $L$, by the fundamental theorem of calculus (Bass, 2013, Theorem 14.16) it holds that, for $p \in [0, 1]$,

$$\mathbb{E}\big[g(m, V, W) - g(p, V, W)\big] = \int_p^m \int_p^\lambda (F' + G')(\vartheta)\, d\vartheta\, d\lambda \le 2L \int_p^m |\lambda - p|\, d\lambda = L|m - p|^2\,.$$

### A.2. Proof of Lemma 2.3

By the bias-variance decomposition:

$$\mathbb{E}\big[|a^\top \hat{\psi}_s - a^\top \psi|^2\big] = \underbrace{\big(\mathbb{E}\big[a^\top \hat{\psi}_s - a^\top \psi\big]\big)^2}_{\text{bias}} + \underbrace{\mathrm{Var}\big[a^\top \hat{\psi}_s\big]}_{\text{variance}}\,.$$

Noting that, for each $t \in \mathbb{N}$, it holds that $\mathbb{E}\big[H_s^\top\big] = f_s^\top \psi$, we have,

$$
\begin{aligned}
\mathbb{E}\big[a^\top \hat{\psi}_t - a^\top \psi\big] &= a^\top (f_s f_s^\top + l^{-1}\mathbf{1}_l)^{-1} f_s f_s^\top \psi \\
&\quad - a^\top (f_s f_s^\top + l^{-1}\mathbf{1}_l)^{-1}(f_s f_s^\top \psi + l^{-1}\psi) \\
&= -a^\top (f_s f_s^\top + l^{-1}\mathbf{1}_l)^{-1} l^{-1}\psi =: (\circ)\,,
\end{aligned}
$$

and hence, by the Cauchy-Schwarz inequality applied to the scalar product $(\alpha, \beta) \mapsto \alpha^\top (f_s f_s^\top + l^{-1} \mathbf{1}_l)^{-1} \beta$, by the fact that $(f_s f_s^\top + l^{-1} \mathbf{1}_l)^{-1} \preceq l^{-1} \mathbf{1}_l^{-1}$ (where, for any two symmetric matrices $A_1, A_2$, we say that $A_1 \preceq A_2$ if and only if $A_2 - A_1$ is semi-positive definite), and by the fact that $\|\psi\|_2^2 \le l$, we can control the bias term as follows

$$
\begin{aligned}
\left(\mathbb{E}[a^\top \hat{\psi}_s - a^\top \psi]\right)^2 &= (\circ)^2 \\
&\le a^\top (f_s f_s^\top + l^{-1} \mathbf{1}_l)^{-1} a \cdot l^{-1} \psi^\top (f_s f_s^\top + l^{-1} \mathbf{1}_l)^{-1} l^{-1} \psi \\
&\le a^\top (f_s f_s^\top + l^{-1} \mathbf{1}_l)^{-1} a \cdot l^{-1} \psi^\top (l^{-1} \mathbf{1}_l)^{-1} l^{-1} \psi \\
&\le a^\top (f_s f_s^\top + l^{-1} \mathbf{1}_l)^{-1} a.
\end{aligned}
$$

Letting $\Delta_s$ be the $s \times s$ diagonal matrix with vector of diagonal elements given by $(\mathrm{Var}[Z_1], \ldots, \mathrm{Var}[Z_s])$, we have

$$
\mathrm{Var}[a^\top \hat{\psi}_s] = a^\top (f_s f_s^\top + l^{-1} \mathbf{1}_l)^{-1} (f_s \Delta_s f_s^\top)(f_s f_s^\top + l^{-1} \mathbf{1}_l)^{-1} a.
$$

Now, given that $Z_1, \ldots, Z_s$ are $[0,1]$-valued, we have that $\Delta_s$ is diagonal with diagonal elements less than 1, and hence $f_s \Delta_s f_s^\top \preceq f_s f_s^\top + l^{-1} \mathbf{1}_l$, which yields a control on the variance term as follows,

$$
\begin{aligned}
\mathrm{Var}[a^\top \hat{\psi}_s] &\le a^\top (f_s f_s^\top + l^{-1} \mathbf{1}_l)^{-1} (f_s f_s^\top + l^{-1} \mathbf{1}_l)(f_s f_s^\top + l^{-1} \mathbf{1}_l)^{-1} a \\
&= a^\top (f_s f_s^\top + l^{-1} \mathbf{1}_l)^{-1} a \ .
\end{aligned}
$$

Putting everything together, we have

$$
\begin{aligned}
\mathbb{E}\left[|a^\top \hat{\psi}_s - a^\top \psi|^2\right] &\le 2 a^\top (f_s f_s^\top + l^{-1} \mathbf{1}_l)^{-1} a = 2 \|a\|_{(f_s f_s^\top + l^{-1} \mathbf{1}_l)^{-1}}^2 \\
&= 2 \|a\|_{(\sum_{r=1}^s a_r a_r^\top + l^{-1} \mathbf{1}_l)^{-1}}^2 \\
&= \left\| \sqrt{2} a \right\|_{(\sum_{r=1}^s a_r a_r^\top + l^{-1} \mathbf{1}_l)^{-1}}^2 \ ,
\end{aligned}
$$

where we recall that for any positive definite matrix $A \in \mathbb{R}^{l \times l}$ and each $u \in \mathbb{R}^l$, we have defined $\|u\|_A \coloneqq \sqrt{u^\top A u}$.

## B. Missing Upper Bound Proofs

In this section, we provide all missing proofs of our regret upper bounds.

### B.1. Proof of Theorem 4.1

Fix any $t \ge 2$. Now, for each $n \in [t-1]$, define $Z_{2n-1} \coloneqq V_n$ and $Z_{2n} \coloneqq W_n$. Define $l \coloneqq d$ and $s \coloneqq 2(t-1)$. For each $n \in [t-1]$, define $a_{2n-1} \coloneqq c_n \eqqcolon a_{2n}$. Let $\psi \coloneqq \phi$ and $a \coloneqq c_t$. Notice that, if $j \in [s]$ is odd, then $\mathbb{E}[Z_j] = \mathbb{E}\left[V_{\frac{j+1}{2}}\right] = c_{\frac{j+1}{2}}^\top \phi = a_j^\top \psi$, while if $j \in [s]$ is even, then $\mathbb{E}[Z_j] = \mathbb{E}\left[W_{\frac{j}{2}}\right] = c_{\frac{j}{2}}^\top \phi = a_j^\top \psi$. Hence, we can apply Lemma 2.3 to obtain

$$
\begin{aligned}
\mathbb{E}\left[|c_t^\top \hat{\phi}_{t-1} - c_t^\top \phi|^2\right] = \mathbb{E}\left[|a^\top \hat{\psi}_s - a^\top \psi|^2\right] &\le \left\| \sqrt{2} a \right\|_{\left(\sum_{j=1}^s a_j a_j^\top + l^{-1} \mathbf{1}_l\right)^{-1}}^2 = \left\| \sqrt{2} c_t \right\|_{\left(2 \sum_{n=1}^{t-1} c_n c_n^\top + d^{-1} \mathbf{1}_d\right)^{-1}}^2 \\
&= \left\| \sqrt{2} c_t \right\|_{\left(\sum_{n=1}^{t-1} (\sqrt{2} c_n)(\sqrt{2} c_n)^\top + d^{-1} \mathbf{1}_d\right)^{-1}}^2 \ .
\end{aligned}
$$

Hence, leveraging Corollary 2.2 and the previous inequality, for any $T \in \mathbb{N}$, we have that

$$
\begin{aligned}
R_T &\le \sum_{t=1}^T 1 \wedge \left( L \mathbb{E}\left[|P_t - c_t^\top \phi|^2\right] \right) \le 1 + \sum_{t=2}^T L \mathbb{E}\left[|c_t^\top \hat{\phi}_{t-1} - c_t^\top \phi|^2\right] \le 1 + \sum_{t=2}^T L \left\| \sqrt{2} c_t \right\|_{\left(\sum_{n=1}^{t-1} (\sqrt{2} c_n)(\sqrt{2} c_n)^\top + d^{-1} \mathbf{1}_d\right)^{-1}}^2 \\
&\le 1 + 2Ld \ln \left( \frac{dd^{-1} + 2d(T-1)}{dd^{-1}} \right) = 1 + 2Ld \ln \left( 1 + 2d(T-1) \right) \le 1 + 2Ld \ln(2dT)
\end{aligned}
$$

where the first inequality of the second line follows from the elliptical potential lemma (Lattimore & Szepesvári, 2020, Lemma 19.4).

If $d < T/2$, this implies that $R_T \le 1 + 2Ld \ln(2dT) \le 1 + 4Ld \ln T$. If, instead, $d \ge T/2$, then, recalling that $L \ge 1$, we obtain once again that $R_T \le T \le 1 + 4Ld \ln T$, concluding the proof.

# C. Missing Lower Bound Proofs

In this section, we provide all the missing proofs of our lower bounds, starting from that of the full-feedback setting.

## C.1. Proof of Theorem 4.2

The high-level idea of this proof is to build a reduction to a non-contextual full-feedback lower bound construction (see, e.g., the one appearing in Bolić et al. 2024, Theorem 5).

Without loss of generality, we assume that $d$ divides $T$. In fact, if we prove the theorem for this case, then, by leveraging that $T \geq 2d$ and $T \geq 4$, the general case follows from

$$R_T \geq bLd \ln\big(\lfloor T/d \rfloor d\big) \geq \frac{b}{2} Ld \ln T .$$

Let $n \coloneqq T/d$. Let $e_1, \ldots, e_d$ be the canonical basis of $\mathbb{R}^d$. Define, for all $i \in [d]$ and $j \in [n]$, the context $c_{j+(i-1)n} \coloneqq e_i$. We assume that these contexts are known to the learner in advance and, therefore, we can restrict the proof to deterministic algorithms without any loss of generality.

Let $L \geq 2$, $J_L \coloneqq \left[\frac{1}{2} - \frac{1}{14L}, \frac{1}{2} + \frac{1}{14L}\right]$, $f \coloneqq \mathbb{I}_{[0,\frac{3}{7}]} + L\mathbb{I}_{J_L} + \mathbb{I}_{[\frac{4}{7},1]}$, and, for any $\varepsilon \in [-1,1]$, $g_\varepsilon \coloneqq -\varepsilon\mathbb{I}_{[\frac{1}{7},\frac{3}{14}]} + \varepsilon\mathbb{I}_{(\frac{3}{14},\frac{2}{7}]}$ and $f_\varepsilon \coloneqq f + g_\varepsilon$. For any $\varepsilon \in [-1,1]$, note that $0 \leq f_\varepsilon \leq L$ and $\int_0^1 f_\varepsilon(x)\,dx = 1$, hence $f_\varepsilon$ is a valid density on $[0,1]$ bounded by $L$. We will denote the corresponding probability measure by $\nu_\varepsilon$, set $\bar{\nu}_\varepsilon \coloneqq \int_{[0,1]} x\,d\nu_\varepsilon(x)$, and notice that direct computations show that $\bar{\nu}_\varepsilon = \frac{1}{2} + \frac{\varepsilon}{196}$. Consider for each $q \in [0,1]$, an i.i.d. sequence $(B_{q,t})_{t\in\mathbb{N}}$ of Bernoulli random variables of parameter $q$, an i.i.d. sequence $(\tilde{B}_t)_{t\in\mathbb{N}}$ of Bernoulli random variables of parameter $1/7$, an i.i.d. sequence $(U_t)_{t\in\mathbb{N}}$ of uniform random variables on $[0,1]$, and uniform random variables $E_1, \ldots, E_d$ on $[-\bar{\varepsilon}_L, \bar{\varepsilon}_L]$, where $\bar{\varepsilon}_L \coloneqq \frac{7}{L}$, such that $\big((B_{q,t})_{t\in\mathbb{N},q\in[0,1]}, (\tilde{B}_t)_{t\in\mathbb{N}}, (U_t)_{t\in\mathbb{N}}, E_1, \ldots, E_d\big)$ is an independent family. Let $\varphi\colon [0,1] \to [0,1]$ be such that, if $U$ is a uniform random variable on $[0,1]$, then the distribution of $\varphi(U)$ has density $\frac{7}{6} \cdot f \cdot \mathbb{I}_{[0,1]\setminus[1/7,2/7]}$ (which exists by the Skorokhod representation theorem (Williams, 1991, Section 17.3)). For each $\varepsilon \in [-1,1]$ and $t \in \mathbb{N}$, define

$$G_{\varepsilon,t} \coloneqq \left(\frac{2+U_t}{14}(1 - B_{\frac{1+\varepsilon}{2},t}) + \frac{3+U_t}{14}B_{\frac{1+\varepsilon}{2},t}\right)\tilde{B}_t + \varphi(U_t)(1 - \tilde{B}_t) , \tag{2}$$

$V_{\varepsilon,t} \coloneqq G_{\varepsilon,2t-1}$, $W_{\varepsilon,t} \coloneqq G_{\varepsilon,2t}$, $\xi_{\varepsilon,t} \coloneqq V_{\varepsilon,t} - \bar{\nu}_\varepsilon$, and $\zeta_{\varepsilon,t} \coloneqq W_{\varepsilon,t} - \bar{\nu}_\varepsilon$. In the following, if $a_1, \ldots, a_d$ is a sequence of elements, we will use the notation $a_{1:d}$ as a shorthand for $(a_1, \ldots, a_d)$. For each $\varepsilon_1, \ldots, \varepsilon_d \in [-1,1]$, each $i \in [d]$, and each $j \in [n]$, define the random variables $\xi^{\varepsilon_{1:d}}_{j+(i-1)n} \coloneqq \xi_{\varepsilon_i, j+(i-1)n}$ and $\zeta^{\varepsilon_{1:d}}_{j+(i-1)n} \coloneqq \zeta_{\varepsilon_i, j+(i-1)n}$. The family $\big(\xi^{\varepsilon_{1:d}}_t, \zeta^{\varepsilon_{1:d}}_t\big)_{t\in[T],\varepsilon_{1:d}\in[-1,1]^d}$ is an independent family, independent of $(E_1, \ldots, E_d)$, and for each $i \in [d]$ and each $j \in [n]$ it can be checked that the two random variables $\xi^{\varepsilon_{1:d}}_{j+(i-1)n}, \zeta^{\varepsilon_{1:d}}_{j+(i-1)n}$ are zero mean with common distribution given by $\nu_{\varepsilon_i}$. For each $\varepsilon_1, \ldots, \varepsilon_d \in [-1,1]$, let $\phi_{\varepsilon_{1:d}} \coloneqq (\bar{\nu}_{\varepsilon_1}, \ldots, \bar{\nu}_{\varepsilon_d})$, and for each $i \in [d]$ and $j \in [n]$, let $V^{\varepsilon_{1:d}}_{j+(i-1)n} \coloneqq c^\top_{j+(i-1)n}\phi_{\varepsilon_{1:d}} + \xi^{\varepsilon_{1:d}}_{j+(i-1)n}$ and $W^{\varepsilon_{1:d}}_{j+(i-1)n} \coloneqq c^\top_{j+(i-1)n}\phi_{\varepsilon_{1:d}} + \zeta^{\varepsilon_{1:d}}_{j+(i-1)n}$. Note that these last two random variables are $[0,1]$-valued zero-mean perturbations of $c^\top_{j+(i-1)n}\phi_{\varepsilon_{1:d}}$ with shared density given by $f_{\varepsilon_i}$, and hence bounded by $L$.

We will show that any algorithm has to suffer the regret inequality in the statement of the theorem if the sequence of evaluations is $V^{\varepsilon_{1:d}}_1, W^{\varepsilon_{1:d}}_1, \ldots, V^{\varepsilon_{1:d}}_T, W^{\varepsilon_{1:d}}_T$, for some $\varepsilon_1, \ldots, \varepsilon_d \in [0,1]$.

Before doing that, we first need the following. For any $\varepsilon_1, \ldots, \varepsilon_d \in [-1,1]$, $p \in [0,1]$, and $t \in [T]$ let $\mathrm{GFT}^{\varepsilon_{1:d}}_t(p) \coloneqq g(p, V^{\varepsilon_{1:d}}_t, W^{\varepsilon_{1:d}}_t)$.

By Lemma 2.1, we have, for all $\varepsilon_1, \ldots, \varepsilon_d \in [-1,1], i \in [d], j \in [n]$, and $p \in [0,1]$,

$$\mathbb{E}\big[\mathrm{GFT}^{\varepsilon_{1:d}}_{j+(i-1)n}(p)\big] = 2\int_0^p \int_0^\lambda f_{\varepsilon_i}(s)\,ds\,d\lambda + 2(\bar{\nu}_{\varepsilon_i} - p)\int_0^p f_{\varepsilon_i}(s)\,ds ,$$

which, together with the fundamental theorem of calculus —(Bass, 2013, Theorem 14.16), noting that $p \mapsto \mathbb{E}\big[\mathrm{GFT}^{\varepsilon_{1:d}}_{j+(i-1)n}(p)\big]$ is absolutely continuous with derivative defined a.e. by $p \mapsto 2(\bar{\nu}_{\varepsilon_i} - p)f_{\varepsilon_i}(p)$— yields, for any $p \in J_L$,

$$\mathbb{E}\big[\mathrm{GFT}^{\varepsilon_{1:d}}_{j+(i-1)n}(\bar{\nu}_{\varepsilon_i})\big] - \mathbb{E}\big[\mathrm{GFT}^{\varepsilon_{1:d}}_{j+(i-1)n}(p)\big] = L|\bar{\nu}_{\varepsilon_i} - p|^2 . \tag{3}$$

Note also that for all $\varepsilon_1, \ldots, \varepsilon_d \in [-\bar{\varepsilon}_L, \bar{\varepsilon}_L]$, $t \in [T]$, and $p \in [0,1] \setminus J_L$, a direct verification shows that

$$\mathbb{E}\big[\mathrm{GFT}^{\varepsilon_{1:d}}_t(p)\big] \leq \mathbb{E}\big[\mathrm{GFT}^{\varepsilon_{1:d}}_t(1/2)\big] . \tag{4}$$

Fix any arbitrary deterministic algorithm for the full feedback setting $(\alpha_t)_{t \in [T]}$, i.e., (given that the contexts $c_1, \ldots, c_T$ are here fixed and declared ahead of time to the learner), a sequence of functions $\alpha_t \colon ([0,1] \times [0,1])^{t-1} \to [0,1]$ mapping past feedback into prices (with the convention that $\alpha_1$ is just a number in $[0,1]$). For each $t \in [T]$, define $\tilde{\alpha}_t \colon ([0,1] \times [0,1])^{t-1} \to J_L$ equal to $\alpha_t$ whenever $\alpha_t$ takes values in $J_L$, and equal to $1/2$ otherwise. Define $Z_1 \coloneqq \frac{1+E_1}{2}, \ldots, Z_d \coloneqq \frac{1+E_d}{2}$.

Now, note the following

$$\sup_{\varepsilon_{1:d} \in [-\bar{\varepsilon}_L, \bar{\varepsilon}_L]^d} \sum_{i=1}^d \sum_{j=1}^n \mathbb{E}\Big[\mathrm{GFT}_{j+(i-1)n}^{\varepsilon_{1:d}}(\bar{\nu}_{\varepsilon_i}) - \mathrm{GFT}_{j+(i-1)n}^{\varepsilon_{1:d}}\big(\alpha_t(V_1^{\varepsilon_{1:d}}, W_1^{\varepsilon_{1:d}}, \ldots, V_{j-1+(i-1)n}^{\varepsilon_{1:d}}, W_{j-1+(i-1)n}^{\varepsilon_{1:d}})\big)\Big]$$

$$\overset{(4)}{\geq} \sup_{\varepsilon_{1:d} \in [-\bar{\varepsilon}_L, \bar{\varepsilon}_L]^d} \sum_{i=1}^d \sum_{j=1}^n \mathbb{E}\Big[\mathrm{GFT}_{j+(i-1)n}^{\varepsilon_{1:d}}(\bar{\nu}_{\varepsilon_i}) - \mathrm{GFT}_{j+(i-1)n}^{\varepsilon_{1:d}}\big(\tilde{\alpha}_t(V_1^{\varepsilon_{1:d}}, W_1^{\varepsilon_{1:d}}, \ldots, V_{j-1+(i-1)n}^{\varepsilon_{1:d}} W_{j-1+(i-1)n}^{\varepsilon_{1:d}})\big)\Big]$$

$$\overset{\spadesuit}{=} L \sup_{\varepsilon_{1:d} \in [-\bar{\varepsilon}_L, \bar{\varepsilon}_L]^d} \sum_{i=1}^d \sum_{j=1}^n \mathbb{E}\Big[\big|\bar{\nu}_{\varepsilon_i} - \tilde{\alpha}_t(V_1^{\varepsilon_{1:d}}, W_1^{\varepsilon_{1:d}}, \ldots, V_{j-1+(i-1)n}^{\varepsilon_{1:d}}, W_{j-1+(i-1)n}^{\varepsilon_{1:d}})\big|^2\Big]$$

$$\geq L \sum_{i=1}^d \sum_{j=1}^n \mathbb{E}\Big[\big|\bar{\nu}_{E_i} - \tilde{\alpha}_t(V_1^{E_{1:d}}, W_1^{E_{1:d}}, \ldots, V_{j-1+(i-1)n}^{E_{1:d}}, W_{j-1+(i-1)n}^{E_{1:d}})\big|^2\Big]$$

$$\overset{\heartsuit}{\geq} L \sum_{i=1}^d \sum_{j=1}^n \mathbb{E}\Big[\big|\bar{\nu}_{E_i} - \mathbb{E}[\bar{\nu}_{E_i} \mid V_1^{E_{1:d}}, W_1^{E_{1:d}}, \ldots, V_{j-1+(i-1)n}^{E_{1:d}}, W_{j-1+(i-1)n}^{E_{1:d}}]\big|^2\Big]$$

$$= \frac{L}{196^2} \sum_{i=1}^d \sum_{j=1}^n \mathbb{E}\Big[\big|E_i - \mathbb{E}[E_i \mid V_1^{E_{1:d}}, W_1^{E_{1:d}} \ldots, V_{j-1+(i-1)n}^{E_{1:d}}, W_{j-1+(i-1)n}^{E_{1:d}}]\big|^2\Big]$$

$$\overset{\blacklozenge}{\geq} \frac{L}{196^2} \sum_{i=1}^d \sum_{j=1}^n \mathbb{E}\Big[\big|E_i - \mathbb{E}[E_i \mid B_{\frac{1+E_i}{2}, 1+2(i-1)n}, \ldots, B_{\frac{1+E_i}{2}, 2(j-1)+2(i-1)n}]\big|^2\Big]$$

$$\overset{\clubsuit}{=} \frac{L}{196^2} \sum_{i=1}^d \sum_{j=1}^n \mathbb{E}\Big[\big|E_i - \mathbb{E}[E_i \mid B_{\frac{1+E_i}{2}, 1}, \ldots, B_{\frac{1+E_i}{2}, 2(j-1)}]\big|^2\Big]$$

$$= \frac{L}{98^2} \sum_{i=1}^d \sum_{j=1}^n \mathbb{E}\Big[\big|Z_i - \mathbb{E}[Z_i \mid B_{Z_i, 1}, \ldots, B_{Z_i, 2(j-1)}]\big|^2\Big]$$

where $\spadesuit$ follows from (3) and the fact that $\tilde{\alpha}_t$ takes values in $J_L$; $\heartsuit$ from the fact that the minimizer of the $L^2(\mathbb{P})$-distance from $\bar{\nu}_{E_i}$ in $\sigma(V_1^{E_{1:d}}, W_1^{E_{1:d}}, \ldots, V_{j-1+(i-1)n}^{E_{1:d}}, W_{j-1+(i-1)n}^{E_{1:d}})$ is $\mathbb{E}[\bar{\nu}_{E_i} \mid V_1^{E_{1:d}}, W_1^{E_{1:d}}, \ldots, V_{j-1+(i-1)n}^{E_{1:d}}, W_{j-1+(i-1)n}^{E_{1:d}}]$ (see, e.g., (Williams, 1991, Section 9.4)); $\blacklozenge$ follows from the fact that, by Equation (2) and the independence of $E_i$ from $\big((B_{q,t})_{t \in \mathbb{N}, q \in [0,1]}, (\tilde{B}_t)_{t \in \mathbb{N}}, (U_t)_{t \in \mathbb{N}}\big)$, the conditional expectation $\mathbb{E}[E_i \mid V_1^{E_{1:d}}, W_1^{E_{1:d}}, \ldots, V_{j-1+(i-1)n}^{E_{1:d}}, W_{j-1+(i-1)n}^{E_{1:d}}]$ is a measurable function of $B_{\frac{1+E_i}{2}, 1+2(i-1)n}, \ldots, B_{\frac{1+E_i}{2}, 2(j-1)+2(i-1)n}$, together with the same observation made in $\heartsuit$ about the minimization of $L^2(\mathbb{P})$ distance; and $\clubsuit$ follows from the fact that the sequence $\big(B_{\frac{1+E_i}{2}, t}\big)_{t \in \mathbb{N}}$ is i.i.d..

Finally, the general term of this last sum is the expected squared distance between the random parameter (drawn uniformly over $[(1-\bar{\varepsilon}_L)/2, (1+\bar{\varepsilon}_L)/2]$) of an i.i.d. sequence of Bernoulli random variables and the conditional expectation of this random parameter given $2(j-1)$ independent realizations of these Bernoullis. A probabilistic argument shows that there exist two universal constants $\tilde{a}, \tilde{b} > 0$ such that, for all $j \geq \tilde{b}L^4$ and each $i \in [d]$,

$$\mathbb{E}\Big[\big|Z_i - \mathbb{E}[Z_i \mid B_{Z_i, 1}, \ldots, B_{Z_i, 2(j-1)}]\big|^2\Big] \geq \tilde{a}\frac{1}{j-1} \ . \tag{5}$$

At a high level, this is because, in an event of probability $\Omega(1)$, if $j$ is large enough, the conditional expectation $\mathbb{E}[Z_i \mid B_{Z_i, 1}, \ldots, B_{Z_i, 2(j-1)}]$ is very close to the empirical average $\frac{1}{2(j-1)} \sum_{s=1}^{2(j-1)} B_{Z_i, s}$, whose expected squared distance from $Z$ is $\Omega(1/(j-1))$. For a formal proof of (5) with explicit constants, we refer the reader to Bolić et al. (2024, Appendix B of the extended arxiv version). Summing over $i \in [d]$ and $j \in [n]$, we obtain that there exist $\varepsilon_1, \ldots, \varepsilon_d \in [-1,1]^d$ such that

$$\sum_{i=1}^d \sum_{j=1}^n \mathbb{E}\Big[\mathrm{GFT}_{j+(i-1)n}^{\varepsilon_{1:d}}(\bar{\nu}_{\varepsilon_i}) - \mathrm{GFT}_{j+(i-1)n}^{\varepsilon_{1:d}}\big(\tilde{\alpha}_t(V_1^{\varepsilon_{1:d}}, W_1^{\varepsilon_{1:d}}, \ldots, V_{j-1+(i-1)n}^{\varepsilon_{1:d}}, W_{j-1+(i-1)n}^{\varepsilon_{1:d}})\big)\Big]$$
$$= \Omega(Ld\ln n) = \Omega(Ld\ln T) \ .$$

## C.2. Proof of Theorem 3.2

The high-level idea of this proof is to build a reduction to a non-contextual lower bound construction (see, e.g., the one appearing in Bolić et al. 2024, Theorem 3).

Fix $L \geq 2$ and $T \in \mathbb{N}$.

We will use the very same notation as in the proof of Theorem 4.2. In particular, the contexts $c_1, \ldots, c_T$ are again the same as before and declared ahead of time to the learner. We will show that for each algorithm for contextual brokerage with 2-bit feedback and each time horizon $T$, if $R_T^{\varepsilon_{1:d}}$ is the regret of the algorithm at time horizon $T$ when the traders' valuations are $V_1^{\varepsilon_{1:d}}, W_1^{\varepsilon_{1:d}}, \ldots, V_T^{\varepsilon_{1:d}}, W_T^{\varepsilon_{1:d}}$, then $\max_{\sigma_{1:d} \in \{-1,1\}^d} R_T^{(\sigma_1 \varepsilon, \ldots, \sigma_d \varepsilon)} = \Omega(\sqrt{dLT})$ if $\varepsilon = \Theta((LT/d)^{-1/4})$ and $T = \Omega(dL^3)$.

Note that for all $\varepsilon_{1:d} \in [-1,1]^d$, $i \in [d]$, $j \in [n]$, and $p < \frac{1}{2}$, if $\varepsilon_i > 0$, then, a direct verification shows that

$$\mathbb{E}\left[\mathrm{GFT}_{j+(i-1)n}^{\varepsilon_{1:d}}(1/2)\right] \geq \mathbb{E}\left[\mathrm{GFT}_{j+(i-1)n}^{\varepsilon_{1:d}}(p)\right]. \tag{6}$$

Similarly, for all $\varepsilon_{1:d} \in [-1,1]^d$, $i \in [d]$, $j \in [n]$, and $p > \frac{1}{2}$, if $\varepsilon_i < 0$, then

$$\mathbb{E}\left[\mathrm{GFT}_{j+(i-1)n}^{\varepsilon_{1:d}}(1/2)\right] \geq \mathbb{E}\left[\mathrm{GFT}_{j+(i-1)n}^{\varepsilon_{1:d}}(p)\right]. \tag{7}$$

Furthermore, a direct verification shows that, for each $\varepsilon_{1:d} \in [-1,1]^d$ and $t \in [T]$,

$$\max_{p \in [0,1]} \mathbb{E}\left[\mathrm{GFT}_t^{\varepsilon_{1:d}}(p)\right] - \max_{p \in [\frac{1}{7}, \frac{2}{7}]} \mathbb{E}\left[\mathrm{GFT}_t^{\varepsilon_{1:d}}(p)\right] \geq \frac{1}{50} = \Omega(1). \tag{8}$$

Now, assume that $T \geq dL^3/14^4$ so that, defining $\varepsilon := (LT/d)^{-1/4}$, we have that for any $\sigma_{1:d} \in \{-1,1\}^d$, any $i \in [d]$ and any $j \in [n]$, the maximizer of the expected gain from trade $p \mapsto \mathbb{E}\left[\mathrm{GFT}_{j+(i-1)n}^{(\sigma_1 \varepsilon, \ldots, \sigma_d \varepsilon)}(p)\right]$ is at $\frac{1}{2} + \frac{\sigma_i \varepsilon}{196}$ and hence belongs to the spike region $J_L$. If $\sigma_i = 1$ (resp., $\sigma_i = -1$), the optimal price for the rounds $1 + (i-1)n, \ldots, in$ belongs to the region $\left(\frac{1}{2}, \frac{1}{2} + \frac{1}{14L}\right]$ (resp., $\left[\frac{1}{2} - \frac{1}{14L}, \frac{1}{2}\right)$). By posting prices in the wrong region $\left[0, \frac{1}{2}\right]$ (resp., $\left[\frac{1}{2}, 1\right]$) in the $\sigma_i = 1$ (resp., $\sigma_i = -1$) case, the learner incurs a $\Omega(L\varepsilon^2) = \Omega(\sqrt{L/dT})$ instantaneous regret by (3) and (6) (resp., (3) and (7)). Then, in order to attempt suffering less than $\Omega(\sqrt{L/T} \cdot n) = \Omega(\sqrt{LT/d})$ regret in the rounds $1 + (i-1)n, \ldots, in$, the algorithm would have to detect the sign of $\sigma_i$ and play accordingly. We will show now that even this strategy will not improve the regret of the algorithm (by more than a constant) because of the cost of determining the sign of $\sigma_i$ with the available feedback. Since for any $i \in [d]$ and $j \in [n]$, the feedback received from the two traders at time $j + (i-1)n$ by posting a price $p$ is $\mathbb{I}\{p \leq V_{j+(i-1)n}^{(\sigma_1 \varepsilon, \ldots, \sigma_d \varepsilon)}\}$ and $\mathbb{I}\{p \leq W_{j+(i-1)n}^{(\sigma_1 \varepsilon, \ldots, \sigma_d \varepsilon)}\}$, the only way to obtain information about (the sign of) $\sigma_i$ is to post in the costly ($\Omega(1)$-instantaneous regret by Equation (8)) sub-optimal region $\left[\frac{1}{7}, \frac{2}{7}\right]$. However, posting prices in the region $\left[\frac{1}{7}, \frac{2}{7}\right]$ at time $j + (i-1)n$ can't give more information about $\sigma_i$ than the information carried by $V_{j+(i-1)n}^{(\sigma_1 \varepsilon, \ldots, \sigma_d \varepsilon)}$ and $W_{j+(i-1)n}^{(\sigma_1 \varepsilon, \ldots, \sigma_d \varepsilon)}$, which, in turn, can't give more information about $\sigma_i$ than the information carried by the two Bernoullis $B_{\frac{1+\sigma_i \varepsilon}{2}, 2(j+(i-1)n)-1}$ and $B_{\frac{1+\sigma_i \varepsilon}{2}, 2(j+(i-1)n)}$. Since only during rounds $1 + (i-1)n, \ldots, in$ is possible to extract information about the sign of $\sigma_i$ and, (via an information-theoretic argument) in order to distinguish the sign of $\sigma_i$ having access to i.i.d. Bernoulli random variables of parameter $\frac{1+\sigma_i \varepsilon}{2}$ requires $\Omega(1/\varepsilon^2) = \Omega(\sqrt{LT/d})$ samples, we are forced to post at least $\Omega(\sqrt{LT/d})$ prices in the costly region $\left[\frac{1}{7}, \frac{2}{7}\right]$ during the rounds $1 + (i-1)n, \ldots, in$ suffering a regret of $\Omega(\sqrt{LT/d}) \cdot \Omega(1) = \Omega(\sqrt{LT/d})$. Putting everything together, no matter what the strategy, each algorithm will pay at least $\Omega(\sqrt{LT/d})$ regret in each epoch $1 + (i-1)n, \ldots, in$ for every $i \in [d]$, resulting in an overall regret of $\Omega(\sqrt{LT/d}) \cdot d = \Omega(\sqrt{dLT})$.

