# OpenReview forum: "A Parametric Contextual Online Learning Theory of Brokerage"
_ICML.cc/2025/Conference — ICML 2025 poster_

### Official Review · Reviewer_2Azh · 2025-02-20

**Overall Recommendation:** 4

**Summary:**

This paper studies brokerage as a contextual online learning problem. Under the assumption that traders' valuations depend linearly on a context available to a broker, the authors design an algorithm achieving a regret bounded by sqrt T. They also derive a corresponding lower bound. The paper then considers the full information feedback, under which an improvement of algorithm 1 leads to a lnT bound on the regret. Here again, a corresponding lower bound is derived. Finally, the authors stress the importance of the bounded density assumption by showing that without this assumption, it is possible to build an instance for which the regret incurred by any algorithm grows linearly with the horizon.

**Claims And Evidence:**

Every claim made in the paper is supported by formal arguments. The authors are transparent regarding the scope of their results.

**Essential References Not Discussed:**

I do not think about any essential reference overlooked by the authors. I would only point out that the sentence “For these reasons, the techniques appearing in contextual linear bandits do not directly translate to our problem.“ (line 216–218) seems a bit too strong. While I understand that gain-from-trade is not linear in the context, it is still piecewise linear, and the proofs in the paper rely a lot on the linear contextual bandit machinery. Likewise, the 2-bits feedback does not look very different from what is considered in duelling and threshold bandits.

**Experimental Designs Or Analyses:**

Not relevant.

**Methods And Evaluation Criteria:**

I fell that experiments are lacking. Given the simplicity of the setting, implementing Algorithm 1 in a simulated environment should not be too challenging. Specifically, comparing the performance of Algorithm 1 in a simulated environment with that of Gaucher (2024) would further support the claim that this method is superior.

**Other Comments Or Suggestions:**

I do not have other comments.

**Other Strengths And Weaknesses:**

strengths:

S1. the paper is very clear and well written. The assumptions and the results are clearly stated. I enjoyed reading this work.

S2. The results are interesting and improve over the known state-of-the-art for this particular problem.

S3. Each upper-bound on regret is supported by a corresponding lower bound, showing that the proposed algorithms achieve almost optimal performances.

 weaknesses:

W1. I believe adding experiments would significantly improve the paper, particularly in supporting the claim that this approach outperforms that of Gaucher et al. (2024). An actual implementation seems even more necessary given that the paper aims to address a practical problem, and the reward environment appears to be relatively straightforward.

W2. The paper makes the underlying assumption that traders are truthful when revealing their willingness to trade to the broker, both in the 2-bits and the full information setting. This seems quite unlikely for actual traders to be so passive and non-strategic. Under-reporting or over-reporting valuations in each period so as to bias the estimate \hat{\phi} seems like a simple strategy to increase gains for traders. This game-theoretic aspect is totally left aside.

**Questions For Authors:**

Q1. Regarding W2, can you discuss why discarding the strategic aspect of your problem makes sense? Ideally, one would want to prove that being truthful is a dominant strategy, or sustainable as a Nash equilibrium. If proving such a result is impossible, it should at least be discussed in the paper.

Q2. As argued in W1, it seems easy to implement an experiment to back the theoretical results of the paper.

EDIT: The authors answered both my questions in a convincing way. I changed my recommendation to Accept.

**Relation To Broader Scientific Literature:**

The literature review is excellent. Even though I have no expertise in the specific brokerage problem, I found it easy to understand how this paper relates to existing studies. In particular, the authors compare their results to the state-of-the-art and clearly explain how their approach improves upon previous work.

**Theoretical Claims:**

I went rapidly through proofs in the main text and I haven’t spotted any blatant problem.

---

> ### Author Rebuttal · Authors · 2025-03-31
>
> We thank the reviewer for their insightful comments.
>
> **Essential References** After reviewing the submission in light of the reviewer remarks, we agree that the mentioned statements could be weakened a bit. We are happy to make the requested changes in the revised version.
>
> **Q1** Great question! One could indeed think of the traders in our model as "impatient". They arrive sequentially and permanently exit if unable to (or after completing a) trade immediately. Under such a sequence of "one-shot" interactions, misreporting valuations offers no future strategic advantage, making truthfulness naturally incentive-compatible. Similar "single-shot participation" modeling assumptions are not uncommon in the bilateral trade literature (Myerson and Satterthwaite, 1983; Blumrosen and Mizrahi, 2016; Colini-Baldeschi et al., 2020; Cesa-Bianchi et al., 2021), as they simplify analyses without sacrificing practical relevance (think of large markets where the broker interacts with new traders every day, like future markets, where the median trader participates in at most 4 trades before leaving the market forever; Ferko et al., Retail Traders in Futures Markets, 2024). Still, the reviewer might wonder:  "Why not even attempt to systematically analyze recurring traders?" The answer is that the problem wouldn't simply become harder, but unlearnable. The proof of Theorem 5.2 implies, as a corollary, that learning is impossible if valuations are chosen strategically. In fact, it gives the even stronger result that learning is impossible even against *oblivious* adversaries (i.e., when valuations are deterministic sequences of numbers in $[0,1]$ that are fixed ahead of time and don't vary as a function of the learner's actions). That said, our model still captures some degree of strategic behavior, as we discuss in our answer to Reviewer AKyE regarding Assumption 1.2.
>
> **Q2**
> Good question.
> Please note that the algorithm SBIP in Gaucher et al. requires additional assumptions to guarantee their stated regret upper bound.
> In particular, the noise distribution for the seller is assumed i.i.d. across times (while we can deal with time-changing noise distributions), and the same is true for the buyer.
> Furthermore, note that an algorithm for the classic bilateral trade problem needs to be adapted to the brokerage setting since, in the brokerage problem, sellers' and buyers' roles are not fixed. A way to do this is by interpreting the agent with the lower valuation as a seller and the agent with the higher valuation as a buyer. Hence setting $S_t = V_t  \land W_t$ and $B_t = V_t \lor W_t$. However, by doing this, the independence between sellers' and buyers' valuations, as well as the assumption that these valuations are zero-mean perturbations of a linear function of the contexts, both present in the statement of the regret guarantees in Gaucher et al., are lost.
>
> This suggests that their algorithm might not yield sublinear guarantees in our setting.
> We validate this intuition by running the experiments the reviewer requested and comparing our regret with theirs.
> As we suspected, their algorithm applied to our setting suffers linear regret, while our Algorithm 1 grows at the theoretical rate of $O(\sqrt{T})$ we proved; see picture in anonymized link: https://i.ibb.co/MkXr19t4/plot.jpg.
>
> The experiments were run for time horizons $T = 1000, 2000, \cdots, 10^4$ for $20$ simulations for each time horizon. At these time steps, the algorithms are tuned by using the parameters prescribed by the respective theories for each specific time horizon, and the corresponding regret is plotted. The dimension $d$ is set to $10$, the noise is uniform in $[-1/4,1/4]$, the unknown vector $\phi$ is picked at random in the $d$-dimensional simplex, while contexts are an i.i.d. process drawn at random in $[1/4,3/4]^d$.
>
> We remain available for clarifications in case we missed something or if something needs further explanation.
> If our response convinces the reviewer, we'd kindly ask them if they would consider adjusting their score accordingly.

---

> > ### Comment · Reviewer_2Azh · 2025-04-01
> >
> > I thank the authors for their high quality answer. Regarding Q1, I am convinced by the "one-shot" interaction setting, which seems practically relevant (Ferko 2024). Regarding Q2, I find that the experiment compelling and sufficient to demonstrate the superiority of the authors' approach. I changed my recommendation to "Accept" accordingly.

---

### Official Review · Reviewer_WcHi · 2025-03-14

**Overall Recommendation:** 4

**Summary:**

This paper considers the brokerage problem between traders for contextual online bilateral trade where in each round, two traders arrive and a context is revealed, then the broker reveals a price, then broker only observes whether the trade with the given price occurred and the identity of buyer and seller. Under some assumptions on (i) the connection between market value and the context, and (ii) the traders' private valuations, they provided the following results
- A linear regret lower bound under no assumption on the density of valuation of traders
- When the density of the valuation is bounded by $L$, they have the following results:
  - For the natural the 2bit information, they provided an algorithm with $\sqrt{LdT\ln{T}}$ regret upper bound and lower bound $\sqrt{Ldt}$ where $d$ is context dimension and $T$ is number of rounds
  - For the case where the broker additionally can observe the private valuations (full information), they provided an algorithm with $O(Ld \ln{T})$ and a matching lower bound.

1. An important conceptual contribution is a structural result in Lemma 2.1, which shows that under some conditions, the optimal price to post is the market value of the item.
2. At high-level, in the 2bit information setting, they provided an exploration-exploitation separated style algorithm that given a context, it either explores by uniformly selecting a price in $[0,1]$ to update the estimated $\hat{\phi}$ or exploits by selecting a price using the estimated $\hat{\phi}$ and context $c_t$, i.e. $P_t = c_t^T \hat{\phi}$. To prove the regret upper bound, they upper bound the number of exploration rounds and the regret in the exploitation rounds.
3. Note that the matching lower bound (up to logarithmic factor) shows that the exploration-exploitation separated algorithm is optimal.
4. Additionally, the lower bound in Section 5, shows that the assumption for bounded density for valuation is needed to achieve no regret in their setting. At a high-level, when the valuation is a probability mass function, then the optimal pricing is not necessarily the market value, which is shown in Example 5.1, and then using this idea, they showed that for a distinct context sequence, $c_1, \ldots, c_T$, it is possible to construct a sequence of valuations for $V_t, W_t$ such that the learner in all rounds is off by a constant from the benchmark because it does not know whether $V_t,W_t$ is either $V_{t,\theta=0}, W_{t,\theta=0}$ or $V_{t,\theta=1}, W_{t,\theta=1}$

**Claims And Evidence:**

Yes

**Essential References Not Discussed:**

I didn't find any

**Experimental Designs Or Analyses:**

NA

**Methods And Evaluation Criteria:**

Yes

**Other Comments Or Suggestions:**

no comments

**Other Strengths And Weaknesses:**

Writing is very clear

**Questions For Authors:**

no questions

**Relation To Broader Scientific Literature:**

Since I'm not familiar with the area, I can't fully assess this, however, given the points mentioned in the paper, I think this could advance the knowledge of online brokerage problems. Especially because the contextual setting is relatively less explored.

**Theoretical Claims:**

- I checked the proof of Theorem 5.2, except those tedious computations mentioned there. I think there is a minor typo in line 407, where $V_{t, \theta}= c_t^T \phi + \xi_t$ where I think it should be $\xi_{t,\theta}$, similarly for $\zeta_t$. But this typo doesn't impact anything.
 - I checked the high level of Theorem 3.1, although I didn't closely verify the part of the proof on the right-hand side of page 6.

---

> ### Author Rebuttal · Authors · 2025-03-31
>
> We thank the reviewer for carefully reading the paper and for their kind words! Thanks also for spotting the typos ($\xi_{t} \rightsquigarrow \xi_{t,\theta}$ and $\zeta_{t} \rightsquigarrow \zeta_{t,\theta}$) on Line  407. We will correct them in the revised version.
>
> It is our understanding that no further clarification or comment is required from us at the moment, but of course, we remain available to provide them upon request.

---

### Official Review · Reviewer_cd62 · 2025-03-14

**Overall Recommendation:** 3

**Summary:**

This paper addresses the problem of sequentially determining transaction prices between two parties based on contextual information. Transactions occur, and rewards are obtained, only when the proposed price falls between the private valuations of the two parties. It is assumed that the expected values of these private valuations can be represented by unknown linear functions of the contextual information. The paper considers two feedback settings: a "2-bit feedback" setting, where only the occurrence of a transaction is observable, and a "full feedback" setting, where the actual valuations are observable. We characterize sufficient conditions to achieve favorable regret bounds, demonstrating that good regret bounds can be attained when the valuation distributions have bounded density functions. Conversely, we also show that, without this assumption, the regret can become linear in the worst case.

**Claims And Evidence:**

The contributions of this paper are theoretical, and all appear to be supported by correct proofs.

**Essential References Not Discussed:**

I am not aware of any particular relevant literature not discussed.

**Experimental Designs Or Analyses:**

N/A

**Methods And Evaluation Criteria:**

The evaluation metric used in this paper (the regret defined in Section 1.1) is natural and reasonable.

**Other Comments Or Suggestions:**

N/A

**Other Strengths And Weaknesses:**

The assumption that noise distributions have bounded densities might seem somewhat unrealistic in practical scenarios. For example, this assumption is not satisfied if valuations follow certain discrete distributions. I view this point as a potential weakness of the paper. Nevertheless, the authors have demonstrated that without this assumption, linear regret can occur in the worst case, thus providing evidence that this assumption is essential.

**Questions For Authors:**

N/A

**Relation To Broader Scientific Literature:**

Although there are several known studies on the online brokerage problem in the context of online learning theory, research addressing contextual settings appears to be relatively scarce.
This paper can be viewed as an extension of existing online brokerage problems to contextual scenarios.
Many aspects of the modeling assumptions, algorithms, and analyses resemble elements found in existing literature on online brokerage problems, online learning, and bandit problems. Essentially, the approach can be viewed as a clever combination of existing techniques.

**Theoretical Claims:**

I checked the correctness of Lemma 2.1, Corollary 2.2, Lemma 2.3, Theorem 3.2, and Theorem 4.1.

---

> ### Author Rebuttal · Authors · 2025-03-31
>
> We thank the reviewer for their review of our work and the comments on our submission.
>
> We are pleased to read that the reviewer evaluates positively the soundness of our setting, the correctness of our results, and our discussion of the relevant related literature.
>
> It is our understanding that no further clarification or comment is required from us at the moment, but of course, we remain available to provide them upon request.

---

### Official Review · Reviewer_AKyE · 2025-03-15

**Overall Recommendation:** 3

**Summary:**

The paper introduces the contextual version of the online brokerage problem. The broker observes a (possibly adversarially generated) context and sets a trading price. The buyer and seller whose private valuations are a perturbed linear function of the context agree for the trade if the price is between the lowest and highest private valuation. The objective function is the gain-from-trade which is the difference between the private valuations subject to the fact that the trade indeed occurs. The goal is to obtain sublinear regret under full feedback and two-bit feedback. The authors propose an algorithm that gets matching regret in both settings under smoothness assumption on the densities of the perturbation. In the unbounded case, the problem becomes unlearnable.

**Claims And Evidence:**

Yes, all the claims are well supported.

**Essential References Not Discussed:**

Connections to the literature on learning how to price in posted price mechanisms and second-price auctions should be discussed more thoroughly. In particular, several works have explored strategic manipulation in similar auction settings:

"Dynamic Incentive-Aware Learning: Robust Pricing in Contextual Auctions" by Negin Golrezaei, Adel Javanmard, and Vahab Mirrokni was published in the Advances in Neural Information Processing Systems 32 (NeurIPS 2019) conference proceedings. This work examines robust pricing strategies in contextual second-price auctions where bidders behave strategically.

Kareem Amin, Afshin Rostamizadeh, and Umar Syed. (2013). Learning prices for repeated auctions with strategic buyers. In Advances in Neural Information Processing Systems, pp. 1169–1177. This paper explores learning strategies in repeated auction settings where buyers act strategically, influencing the pricing dynamics.

There are also earlier papers that studied learning in auction settings but did not account for strategic behavior. It would be helpful to clarify whether the authors borrowed any ideas or insights from this existing literature. While there are clear differences between the settings, the nature of the feedback in both cases is similar, which raises the possibility that techniques from the earlier work could have informed the proposed approach. Addressing these connections would strengthen the positioning of the paper within the broader research landscape.

**Experimental Designs Or Analyses:**

The paper has no experimental section.

**Methods And Evaluation Criteria:**

Yes.

**Other Comments Or Suggestions:**

NA

**Other Strengths And Weaknesses:**

**Strengths:**
The learning algorithms and its analysis are quite neat and intuitive.

The authors obtain tight regret bounds for all the considered settings (up to log terms).

**Weaknesses:**
The assumptions are possibly too strong. See details in "questions for authors" section.

**Questions For Authors:**

The proofs in the paper heavily rely on the two main assumptions. **Assumption 1.2**, in particular, raises some questions about its realism. This assumption states that traders' private valuations are zero-mean perturbations of the market value. However, if the market prices (which serve as the context) are adversarially generated, it seems inconsistent to assume that the traders' valuations would remain unbiased and follow a simple random pattern around the market price. In real-world financial markets, traders' valuations are influenced by complex factors such as market manipulation, asymmetric information, strategic behavior, and behavioral biases — all of which could introduce systematic deviations from the market price rather than just zero-mean noise.

Furthermore, adversarial market prices suggest that the market environment itself is highly unpredictable and potentially influenced by strategic forces. Under such conditions, it seems natural to expect that traders' valuations would also reflect some degree of strategic adaptation rather than simple random perturbations. For example, traders might adjust their valuations based on market trends, competitor behavior, or even attempts to exploit the broker's algorithm. Therefore, it would be helpful if the authors could clarify the rationale behind this assumption and discuss whether relaxing it — for instance, allowing for adversarial or biased deviations in valuations — would significantly affect the theoretical results. Additionally, providing empirical or theoretical justification for why the perturbation model remains reasonable under adversarial contexts would strengthen the validity of the assumption and enhance the practical relevance of the model.

2- For a fixed $ \phi \in [0, 1]^d %, if the contexts $c_t \in [0, 1]^d $ are allowed to be adversarially generated, it’s not immediately clear how the inner product
$m_t = \langle \phi, c_t \rangle$ would stay within the bounded range of \([0, 1]\). **The range of the inner product  $\langle \phi, c_t \rangle $ is theoretically $[0, d]$ because both $ \phi $ and $ c_t $ are vectors of dimension $ d $ with entries in $[0, 1]$. **In the worst-case scenario, if all components of \( \phi \) and \( c_t \) are maximized at 1, the inner product would reach its upper bound of  $d $, which exceeds the target range of $[0, 1]$.

This raises two key questions:

1. **Bounding the Market Value:** If the goal is to model $m_t$ as a valid market value within $[0, 1]$, how is the inner product restricted to stay in this range when the contexts are adversarially generated? If the adversarial generation of contexts is unconstrained, the inner product could easily exceed 1.

2. **Impact on Learning and Regret Bounds:** If the inner product can exceed 1, the broker’s learning strategy and regret guarantees might break down, since the reward function and the regret definition are based on the assumption that the market value is bounded within $[0, 1]$. If the true range is wider, the theoretical analysis might require adjustments or additional constraints on the adversarial nature of the contexts.

One way to address this could be to explicitly impose a normalization or projection step that ensures the computed market value remains in the interval $[0, 1]$ — for example, defining the market value as
$
m_t = \min(1, \max(0, \langle \phi, c_t \rangle)).
$

3- Moreover, what are the assumptions on the random variables $\xi$ and $\zeta$ to ensure that the private valuations of the sellers and buyers are in $[0, 1]$?

**Relation To Broader Scientific Literature:**

The paper sheds light on the contextual version of the online brokerage problem which is practically relevant (as brokers often have external information before setting the price). The learning algorithms in the paper are quite novel to the best of my knowledge. In the 2-bit feedback model, the algorithm obtains better regret bounds (sqrt(T) vs T^⅔) than the most recent work in this area [GBC+24] under a more relaxed setting which highlights the contributions of the paper. Although constructing the lower bound instance is quite intricate and involved, the paper mostly leverages the existing work of [BCR24].

[GBC+24] Gaucher, S., Bernasconi, M., Castiglioni, M., Celli, A., & Perchet, V. (2024). Feature-based online bilateral trade. arXiv preprint arXiv:2405.18183.

**Theoretical Claims:**

Yes, I checked the proofs of all the theoretical claims and they seem fine.

---

> ### Author Rebuttal · Authors · 2025-03-31
>
> We thank the reviewer for their insightful comments.
>
> **Additional references** We thank the reviewer for bringing the two additional references to our attention. We will add them to the revised version. Regarding one-sided problems (like auctions or dynamic pricing), no techniques we are aware of can be directly applied to or give clear insights into two-sided problems (like our brokerage problem) other than high-level online learning ideas (like constructing hard lower-bound instances by hiding slightly better actions in an appropriately constructed set of "hard-to-distinguish" actions). If the reviewer has any specific references in mind, we are happy to add them to the revised related works section. If they want us to add a concise part on related one-sided problems (like auctions or dynamic pricing), we can do that too.
>
> With **Assumption 1.2**, we aimed to capture the variability of individual preferences rather than systematic strategic behavior around a target quantity. The assumption is consistent with economic models where the "market price" is the notion that aggregates strategic behaviors, asymmetric information, and biases, so that deviations from the market price reduce to *unbiased* residual noise once strategic behaviors are reflected in the market price itself. In other words, the market price does not represent an asset's inherent value but the market participants' average opinion. Importantly, our theoretical results do not require identical distributions across time. The market's opinion can vary arbitrarily (even adversarially) over time, and the noise distributions too (representing periods where opinions are more or less aligned). The only concept that remains stationary is that the market participants' average opinion determines assets' "market values".
>
> We agree that altering this assumption by allowing systematic biases or strategic deviations around a notion of "inherent value" would be an interesting alternative path. We will mention this future research direction in the conclusions section and thank the reviewer for raising this point!
>
> **Assumption 1.1 (Market values and contexts)** is indeed an assumption on market values *and* contexts. A natural and common special case in which this assumption holds is when $|| \phi ||_1 = 1$, i.e., when $\phi$ is an unknown vector of weights belonging to the probability simplex, representing how important each component of the context vector $c_t$ is to determine the market value at round $t$ (with the boundary cases being $\phi_i = \mathbb{I}$ {$i=j$}, for some $j$, i.e., all the information necessary to reconstruct the market value is contained in a component $j$, or $\phi_i = 1/d$ for all $i$, i.e., all components are equally as informative). In this case, $\langle \phi, c_t \rangle \in [0,1]$ for all $t$. By Holder's inequality, this can be further generalized by assuming that $||\phi||_p \le 1$ and $||c_t||_q \le 1$ (where $q$ is the Holder conjugate of $p$), which is just another way of expressing boundedness of the vectors. Instead of fixing one of these specific assumptions, we opted for the most general case where only the property $\langle \phi, c_t \rangle \in [0,1]$ is assumed.
> Note that this assumption is merely for the sake of consistency (it is intuitive to model valuations and market prices all belonging to the same set $[0,1]$), but the same algorithm works, and the same analysis yields the same rate (up to a factor of $d$) if market prices are in $[0,d]$.
> Thus, our core theoretical insights remain intact up to minor technical adjustments if this assumption is lifted.
>
> **Assumption 1.2: $V_t, W_t \in [0,1]$**
> For $V_t = m_t + \xi_t$ and $W_t = m_t +\zeta_t$ to be bounded in $[0,1]$ for all $t$, the noise random variables $\xi_t$ and $\zeta_t$ need to be bounded in $[- m_t , 1 - m_t]$. A sufficient condition for this to happen is to assume that $m_t \in [a,b]$, with $0<a<b<1$. This is simply stating that the value of the assets traded is never 0 (in real life, if the asset is, e.g., some stock of a company, then $m_t = 0$ only if the company goes bankrupt, in which case, of course, people would not trade the stock) or 1 (which is equally as natural since the fact that prices are normalized in $[0,1]$ means that $1$ represents an upper bound on the largest amount of money that traders would spend to exchange an asset). Note that this boundedness condition does not conflict with the zero-mean assumption but simply ensures that valuations are always interpretable as prices within the normalized range $[0,1]$.
>
> If, in light of this discussion, the reviewer agrees that the assumptions we made (Assumptions 1.1 and 1.2) do not significantly restrict the general applicability of our theory, we'd kindly ask them if they would consider adjusting their score accordingly. We remain available for clarifications in case we missed something or if something needs further explanation.

---

### Decision · Program_Chairs · 2025-05-01

**Decision:**

Accept (poster)

**Comment:**

This paper studies the problem of learning in a repeated bilateral trade setting --- a topic that has gained traction in the recent few years. The model studied in this paper is a nice twist to what's been studied recently, namely, that either party can be a buyer or a seller. At each step $t$, two agents arrive, with private valuations $V_t$ and $W_t$, and a broker (or mechanism designer) is allowed to post a price $p_t$. If $p_t$ happens to fall in between $V_t$ and $W_t$ a trade happens, regardless of whether $V_t$ is larger or $W_t$  is larger. The valuations $V_t$ and $W_t$ are derived from a market value $m_t$ plus two (possibly correlated) zero-mean variables $\zeta_t$ and $\xi_t$, with densities bounded by $L$. I.e., $V_t = m_t + \xi_t$ and $W_t = m_t + \zeta_t$, and the market value $m_t$ itself is related to a $d$-dimensional context $c_t$ (the context is revealed to the mechanism designer) via a linear relationship $m_t = \langle c_t, \phi \rangle$. The mechanism designer seeks to minimize regret in gains from trade, namely, the difference between the gains from trade obtained by the optimal-in-hindsight function mapping contexts to prices, to the gains from trade obtained by the broker. The broker's feedback is just two bits: after posting their price $P_t$, the broker can observe whether the price is larger or smaller than $V_t$ and likewise for $W_t$.

The committee appreciated several aspects of the paper including (a) tight regret bounds as a function of all quantities unvolved, namely, the density bound $L$, dimension $d$ and time horizon $T$, (b) a clear separation between the full feedback setting (where $V_t$ and $W_t$ are revealed after each round $t$) and the one studied in this paper, and (c) the problem itself being a nice twist to the regular bilateral trade setting.

The paper is a solid contribution to ICML.